# Coordination chemogenetics for activation of GPCR-type glutamate receptors in brain tissue

Kento Ojima[1,2,6], Wataru Kakegawa[3,6], Tokiwa Yamasaki[3], Yuta Miura[1], Masayuki Itoh[3], Yukiko Michibata[2], Ryou Kubota [2], Tomohiro Doura[1], Eriko Miura[3], Hiroshi Nonaka [2], Seiya Mizuno[4], Satoru Takahashi [4], Michisuke Yuzaki [3✉], Itaru Hamachi [2✉] & Shigeki Kiyonaka [1,5✉]

Direct activation of cell-surface receptors is highly desirable for elucidating their physiological roles. A potential approach for cell-type-specific activation of a receptor subtype is chemogenetics, in which both point mutagenesis of the receptors and designed ligands are used. However, ligand-binding properties are affected in most cases. Here, we developed a chemogenetic method for direct activation of metabotropic glutamate receptor 1 (mGlu1), which plays essential roles in cerebellar functions in the brain. Our screening identified a mGlu1 mutant, mGlu1(N264H), that was activated directly by palladium complexes. A palladium complex showing low cytotoxicity successfully activated mGlu1 in mGlu1(N264H) knock-in mice, revealing that activation of endogenous mGlu1 is sufficient to evoke the critical cellular mechanism of synaptic plasticity, a basis of motor learning in the cerebellum. Moreover, cell-type-specific activation of mGlu1 was demonstrated successfully using adeno-associated viruses in mice, which shows the potential utility of this chemogenetics for clarifying the physiological roles of mGlu1 in a cell-type-specific manner.

[1] Department of Biomolecular Engineering, Graduate School of Engineering, Nagoya University, Nagoya 464-8603, Japan. [2] Department of Synthetic Chemistry and Biological Chemistry, Graduate School of Engineering, Kyoto University, Kyoto 615-8510, Japan. [3] Department of Neurophysiology, Keio University School of Medicine, Tokyo 160-8582, Japan. [4] Laboratory Animal Resource Center in Transborder Medical Research Center, Faculty of Medicine, University of Tsukuba, Tsukuba 305-8575, Japan. [5] Institute of Nano-Life-Systems, Institutes of Innovation for Future Society, Nagoya University, Nagoya 464-8603, Japan. [6]These authors contributed equally: Kento Ojima, Wataru Kakegawa. ✉email: myuzaki@keio.jp; ihamachi@sbchem.kyoto-u.ac.jp; kiyonaka@chembio.nagoya-u.ac.jp

Cell-surface receptors have indispensable roles in transmitting extracellular information into cells. Most receptors belong to protein families composed of highly homologous subtypes, each with unique physiological functions. In elucidating the roles of each receptor subtype, it is highly desirable to develop activators that selectively target a particular subtype. However, the high homology of the orthosteric ligand-binding site hampers the development of subtype-selective agonists[1,2]. Instead, researchers have focused on positive allosteric modulators (PAMs), which increase the efficacy of endogenous ligands by binding to an allosteric site as subtype-selective ligands. Although PAMs represent potentially powerful therapeutic agents to increase the efficacy of receptors in situations where the concentration of endogenous ligands is reduced, researchers should be careful when using PAMs to clarify the physiological functions of receptors. Given the non-uniform concentration of endogenous ligands such as neurotransmitters in tissues[3], the effects of a PAM may reflect the concentration of the ligands rather than the receptor function. Thus, direct activators with high subtype-selectivity are ideal for understanding the physiological roles of the target receptor.

A potential method for direct activation of target receptors is chemogenetics, in which proteins are genetically engineered to interact with the designed ligand selectively[4]. As pioneering research, a κ-opioid receptor mutant, which abrogated affinity to the natural ligand yet preserved stimulation by an artificial ligand, has been reported[5]. As improved approaches, designer receptors exclusively activated by designer drugs (DREADDs), which have low constitutive activity and high selectivity to an artificial ligand, clozapine-N-oxide, have been developed by directed molecular evolution of muscarinic acetylcholine receptors[6]. In association with improving the designed ligands, DREADDs are powerful tools to control cellular signaling of target cells in tissues or living animals[7,8]. For ligand-gated ion channels, engineered receptors (PSAMs) and designed agonists (PSEMs) pairs have been reported using the bump-and-hole strategy[9,10]. Although powerful for manipulating cellular signals, these methods are unsuitable for characterizing the physiological roles of each receptor subtype because engineered receptors lacking affinity to endogenous ligands need to be ectopically expressed.

Thus, developing new chemogenetic methods to directly activate receptor subtypes with synthetic ligands without affecting the original receptor function is highly desirable. As a potential approach, we have recently reported a chemogenetic method using coordination chemistry for the AMPA receptor (AMPAR) and metabotropic glutamate receptor 1 (mGlu1), which belong to ion-channel-type (ionotropic) and G protein-coupled-type (metabotropic) glutamate receptors, respectively[11,12]. In this method, structure-based incorporation of coordinating amino acid residues as allosteric sites into glutamate receptors allowed metal-complex-associated activation of glutamate receptors in HEK293 cells or cultured neurons. Although useful for chemogenetic activation of these receptors, the chemogenetic coordination acted as a PAM, which increased the affinity of glutamate to the receptors.

In this report, we developed a direct activation method of mGlu1 by chemogenetics, termed dA-CBC (direct-activation via coordination-based chemogenetics). In the engineered receptor, a metal-complex [(Pd(bpy), Pd(bpy)(NO$_3$)$_2$; bpy, 2,2'-bipyridine)] activated the mGlu1 mutant directly without potentiating the affinity of glutamate. We also designed a derivative of Pd(bpy) with low toxicity to neurons. In mice with the mGlu1 mutation, the newly developed palladium complex successfully induced downstream signals of mGlu1 in acute cerebellar slices of the mice. Notably, the palladium complex evoked synaptic plasticity such as long-term depression (LTD), a putative cellular model of information storage for motor learning in the cerebellum of mammals. Moreover, the dA-CBC strategy was applied successfully to cell-type-specific activation of mGlu1 using adeno-associated viruses (AAVs) encoding the mGlu1 mutant. Thus, dA-CBC represents a powerful method for understanding the physiological roles of mGlu1 in brain tissue.

## Results

**Screening the direct-activator-type mGlu1 mutant by dA-CBC.** mGlu1, with a large extracellular and a 7-transmembrane (7TM) domain, belongs to the class C G-protein-coupled receptor (GPCR) family. The extracellular domain is composed of a Venus Flytrap (VFT) domain and a cysteine-rich domain (CRD) for glutamate binding and as a semi-rigid linker, respectively (Fig. 1a)[13]. Glutamate binding to the VFT domain induces closure of the domain in the mGlu1 homodimer, which is transmitted to the 7TM domain via CRD for receptor activation (Fig. 1a). We reported chemogenetic methods previously for allosteric activation of glutamate receptors by regulating the open and closed form of the VFT domain using metal coordination, termed on-cell coordination chemistry (OcCC). We identified a mGlu1 mutant, mGlu1(P58H/N264H), whose glutamate-induced responses were allosterically activated by Pd(bpy) (Fig. 1b, top). In the mutant, 1 μM of glutamate, which is insufficient for receptor activation, successfully induced intracellular signals in the presence of Pd(bpy). Importantly, the mGlu1 mutant was activated by Pd(bpy) even in the absence of glutamate, which guided our hypothesis that the introduction of histidine mutations in appropriate positions should induce direct activation of the mGlu1 by metal coordination without potentiating the affinity of glutamate (Fig. 1b, bottom). This strategy is termed dA-CBC, and our previous OcCC is renamed A-CBC (allosteric activation via coordination-based chemogenetics).

Here, we prepared seven mGlu1 mutants with two histidine mutations on both upper and lower lobes at the entrance of the VFT domain (Fig. 1c). The mutation sites, which are not involved in glutamate binding, were selected to minimize the influence on receptor function and based on structural information available for open (i.e., apo) and closed (i.e., glutamate-binding) conformations[14,15]. We hypothesized that domain closure occurs by metal coordination instead of glutamate binding when the mutated histidines work as a bidentate chelator for metal ions or complexes. mGlu1 is a Gq-coupled GPCR, which elevates the intracellular Ca$^{2+}$ concentration ([Ca$^{2+}$]$_i$) via phospholipase C (PLC) activation. Receptor activation was evaluated by monitoring [Ca$^{2+}$]$_i$ changes using a fluorescent Ca$^{2+}$ indicator, Fura-2, after treatment with 10 μM metal ions (Ni$^{2+}$, Cu$^{2+}$, Zn$^{2+}$, Cd$^{2+}$, or Pd$^{2+}$) or a metal complex (Pd(bpy)) in HEK293 cells transfected with each mGlu1 mutant (Supplementary Fig. 1a). The 1st screening identified five mutants (P57H/N264H, P58H/N264H, A59H/N264H, E60H/N264H, and K61H/N264H) that were prominently activated by Pd(bpy) (Fig. 1d, e and Supplementary Fig. 1b). Although Pd$^{2+}$ also activated three mutants (P58H/N264H, E60H/N264H, and K61H/N264H), the obtained Δratio values were not that high when compared with those by Pd(bpy). Other metal ions (Ni$^{2+}$, Cu$^{2+}$, Zn$^{2+}$ or Cd$^{2+}$) failed to activate these mGlu1 mutants. We selected three mutants (P58H/N264H, A59H/N264H, and K61H/N264H) as hit mutants from the 1st screening because these three mutants showed intact glutamate-induced responses and prominent Pd(bpy)-induced responses (Fig. 1e). The calculated half-maximal effective concentration (EC$_{50}$) values of Pd(bpy) were 1.2–3.8 μM for the three mutants (Fig. 1f). The dose-dependency of glutamate-evoked responses was not affected significantly in these mutants when compared with that of wild-type (WT) mGlu1 (Supplementary Fig. 2), indicating that introduction of these histidine mutations did not affect the original receptor function of mGlu1.

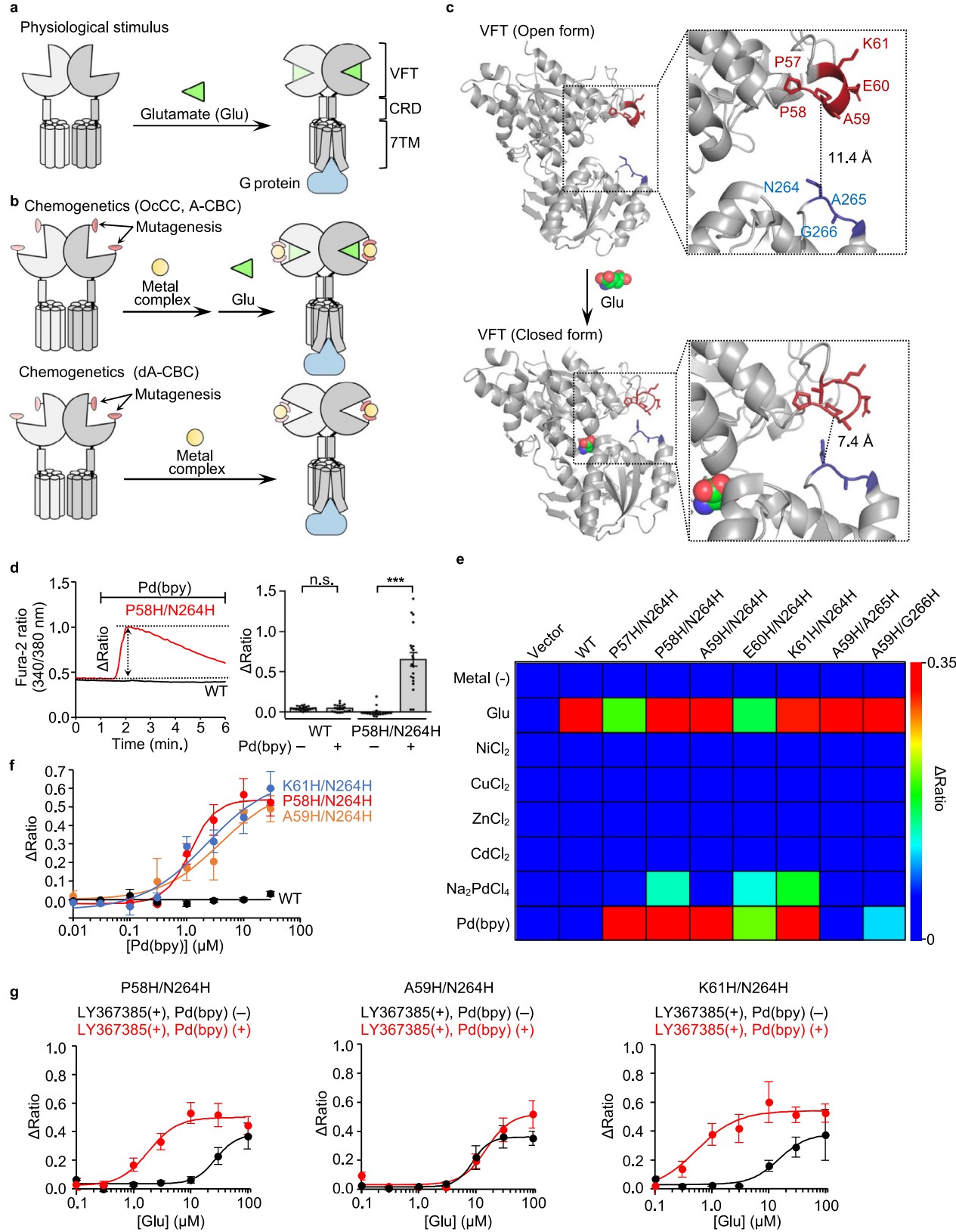

Although hit mutants and Pd(bpy) pairs were obtained, it was unclear whether Pd(bpy) acts as a direct activator or an activator coupled with PAM activity (i.e., ago-PAM)[2] for the mGlu1 mutants. Then, in the next step (i.e., 2nd screening), we aimed to find mGlu1 mutants activated by Pd(bpy) directly without potentiating the affinity of glutamate. To evaluate the PAM effect selectively by

excluding the direct activation effect of Pd(bpy), we used a mGlu1-selective orthosteric antagonist, LY367385 (ref. [16]), which competitively binds with glutamate to the VFT domain (Supplementary Fig. 3a). As shown in Supplementary Figures 3b, c, pretreatment with LY367385 suppressed glutamate-induced responses. Importantly, 10 μM LY367385 also inhibited Pd(bpy)-induced Ca²⁺

**Fig. 1 Screening of the direct-activation-type mGlu1 mutants. a** Schematic illustration of glutamate-induced conformational changes of mGlu1.
**b** Schematic illustration of the coordination-based chemogenetics (CBC). Upper: allosteric activation of mGlu1 by A-CBC (OcCC in our previous terminology). Bottom: direct activation of mGlu1 by dA-CBC. **c** Conformational change of the VFT on the glutamate binding. Upper: open form in the apo state (Protein Data Bank (PDB) 1EWT). Lower: closed form in the glutamate-binding state (PDB 1EWK). The mutation sites at the upper and lower lobes are highlighted in red and blue. **d** Pd(bpy)-induced mGlu1 responses in HEK293 cells. Left: representative trace of $Ca^{2+}$ response induced by 10 μM of Pd(bpy) in HEK293 cells transfected with the plasmid of mGlu1 P58H/N264H mutant (red) or WT mGlu1 (black). The Δratio is defined as the difference between the maximum and the initial ratio values. Right: averaged Δratio in the presence or absence of 10 μM of Pd(bpy) for WT ($P = 0.9423$) and P58H/N264H ($P = 2.214 \times 10^{-7}$). ($n = 20$). (Two-tailed Welch's $t$-test, ***$P < 0.001$, n.s. not significant). **e** The heatmap shows the averaged Δratio induced by 10 μM glutamate, metal ions, or a metal complex. See Supplementary Fig. 1b for each data. **f** Concentration-dependent curves for Pd(bpy) in HEK293 cells expressing mGlu1 WT (black), P58H/N264H (red), A59H/N264H (orange), or K61H/N264H (blue). ($n = 20$). $EC_{50}$ values were 1.2, 3.8, and 2.3 μM for P58H/N264H, A59H/N264H, and K61H/N264H, respectively. **g** Evaluation of the positive allosteric effect of Pd(bpy) to the mGlu1 mutants. Effects of 3 μM Pd(bpy) on the concentration-dependency of the glutamate-responses were examined in the presence of 10 μM LY367385 in HEK293 cells expressing the hit mutants (P58H/N264H, A59H/N264H, K61H/N264H). ($n = 20$). Data are presented as mean ± s.e.m. See Supplementary Figure 3g, h for the representative traces.

responses in the mGlu1(P58H/N264H) mutant (Supplementary Fig. 3d, e), which showed that direct activation of Pd(bpy) can be suppressed by the orthosteric antagonist. Then, we directly evaluated the positive allosteric effect of Pd(bpy) for the glutamate-induced response by pretreatment with LY367385 (Supplementary Fig. 3f). The dose-dependent glutamate-induced responses of the hit mutants were obtained even in the presence of LY367385, and the positive allosteric effect of Pd(bpy) was largely different among these mutants (Fig. 1g and Supplementary Fig. 3g, h). The $EC_{50}$ values of glutamate were shifted to the left for the P58H/N264H or K61H/N264H mutant (Fig. 1g). In contrast, a prominent shift of the $EC_{50}$ value was not observed for the A59H/N264H mutant, indicating that the A59H/N264H mutant was activated by Pd(bpy) without potentiating glutamate affinity. In contrast, Pd(bpy) acted as ago-PAM for P58H/N264H and K61H/N264H mutants. Thus, the screening results identified the A59H/N264H mutant and Pd(bpy) as a hit pair for dA-CBC of mGlu1.

**Direct activation mechanism of the mGlu1 mutant by dA-CBC.** We next investigated the activation mechanism of the mGlu1 A59H/N264H mutant. The corresponding single mutants (A59H and N264H mutants) were prepared to examine the bidentate coordination of Pd(bpy) to both histidine residues. As shown in Fig. 2a, Pd(bpy) failed to activate the A59H mutant. In contrast, the N264H mutant was unexpectedly activated by Pd(bpy). In both mutants, glutamate responses were intact, suggesting that receptor function was unaffected by the mutations (Supplementary Fig. 4). Regarding other Pd(bpy)-responsive mutants (P58H/N264H and K61H/N264H), the corresponding single mutants (P58H and K61H) failed to be activated by Pd(bpy) (Fig. 2a). These results indicate that the single N264H mutation is sufficient for the chemogenetic activation of mGlu1.

Considering the putative activation mechanism of dA-CBC (Fig. 1b, bottom), a coordination partner of the N264H mutation for Pd(bpy) binding is required on the upper lobe of the VFT domain. We found four coordinating amino acid residues (H54, H55, E60, and H111) within 15 Å from the N264 residue in the crystal structure of the closed form of the VFT domain (Fig. 2b and Supplementary Table 1). These candidate amino acids were exchanged to phenylalanine or alanine in the N264H mutant, and Pd(bpy)-induced responses were examined for these mutants. As shown in Fig. 2c, the H54F/N264H and H55F/N264H mutants failed to be activated by Pd(bpy). However, glutamate-induced responses were also lost in the H54F/N264H mutant but not the H55F/N264H mutant. Given that H54 interacts with D107 in the crystal structure[14], substituting this residue is likely to be lethal for receptor activity in the H54F/N264H mutant. Collectively, although the involvement of H54 cannot be excluded, endogenous H55 is probably the coordination partner of the N264H mutation

for Pd(bpy)-induced chemogenetic activation. The $EC_{50}$ value of Pd(bpy) for the N264H mutant was 1.6 μM, and glutamate-induced responses were unaffected by the N264H mutation (Fig. 3a, b). Importantly, the dose-dependency of the glutamate-induced responses in the presence of LY367385 was unaffected in the presence of Pd(bpy), indicating that the affinity of glutamate was not potentiated by Pd(bpy) in the mGlu1(N264H) mutant (Fig. 3c). Thus, the single point mutation (N264H) is sufficient for the direct activation of mGlu1 by Pd(bpy).

The chemogenetic activation of the mGlu1(N264H) mutant in HEK293 cells was examined in more detail. Western blotting indicated that the total expression level of the N264H mutant was comparable to that of WT mGlu1 (Fig. 3d). A newly designed mGlu1-selective fluorescent probe, FITM-Cy3, was designed to quantify the expression level of mGlu1 on the cell surface under live-cell conditions (Fig. 3e). FITM, a negative allosteric modulator (NAM) of mGlu1 which binds to the 7TM domain[17], was conjugated with an anionic fluorescent dye, sulfo-Cy3, to obtain FITM-Cy3. Confocal live imaging clearly showed that FITM-Cy3 visualizes surface mGlu1 with a dissociation constant ($K_d$) of 6.8 nM in HEK293 cells (Supplementary Fig. 5), and FITM-Cy3 clarified that surface expression of the N264H mutant was comparable to that of WT mGlu1 (Fig. 3f).

The time required for Pd(bpy)-induced mGlu1 activation to reach a peak ratio was longer when compared with that of glutamate-induced activation ($58.0 \pm 8.5$ s or $25.7 \pm 1.9$ s for Pd(bpy) or glutamate, respectively) (Supplementary Fig. 6). The kinetics of the wash-out of Pd(bpy) was also slower than that of glutamate ($49.0 \pm 6.4$ s or $22.0 \pm 3.2$ s for Pd(bpy) or glutamate, respectively). These observations indicate that the binding kinetics are slightly different between Pd(bpy) and glutamate. However, repeated ratio increases were observed after washing out Pd(bpy) in the culture medium (Fig. 3g), showing the reversible action of Pd(bpy) for direct activation of mGlu1.

We also examined the effects of Pd(bpy) on mGlu1-dependent Gq signaling. The Pd(bpy)-induced $Ca^{2+}$ responses were inhibited by LY367385 or FITM, a competitive or non-competitive antagonist of mGlu1, respectively (Supplementary Fig. 7a, b). The $Ca^{2+}$ responses were also suppressed by YM-254890, a Gq-selective inhibitor, and U73122, a PLC inhibitor (Supplementary Fig. 7c, d), indicating that Pd(bpy) activates the conventional Gq-pathway of mGlu1. Consistently, inositol 1,4,5-triphosphate (IP3) production was increased by Pd(bpy) in HEK293 cells transfected with mGlu1(N264H), which was evaluated by the accumulation of inositol 1-phosphate (IP1), the degradation product of IP3. (Supplementary Fig. 8). Thus, we can directly and reversibly activate mGlu1 and its downstream signal pathways with minimal disturbance to receptor function by using Pd(bpy) and a single-point mutation (N264H) of mGlu1 in the dA-CBC method.

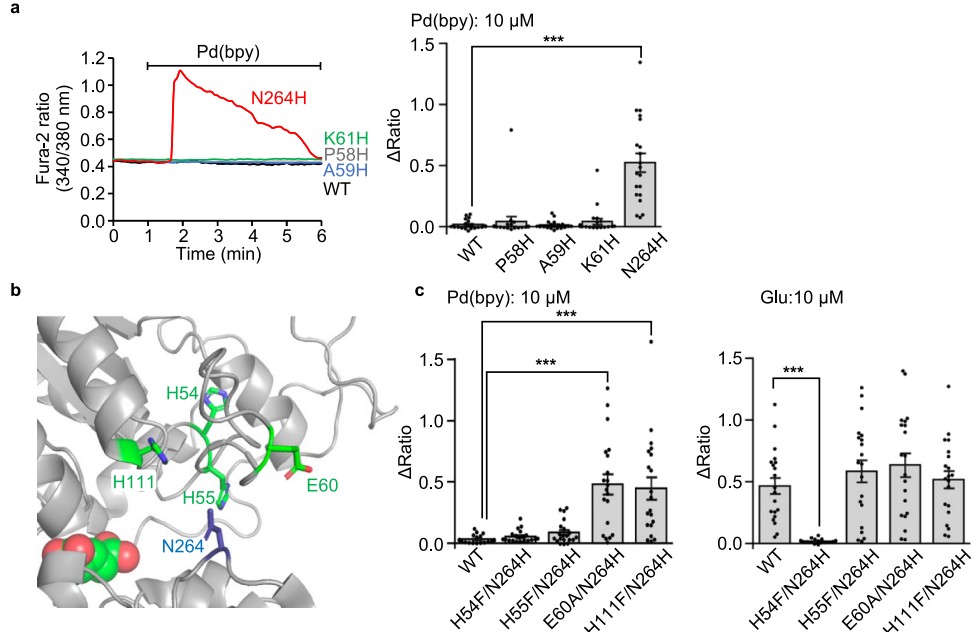

**Fig. 2 Direct activation mechanism of of the mGlu1 mutant by dA-CBC. a** Pd(bpy)-induced mGlu1 responses in HEK293 cells expressing the mGlu1 mutant. Left: representative traces of $Ca^{2+}$ responses induced by 10 μM of Pd(bpy) in HEK293 cells expressed with the mGlu1 N264H (red), P58H (gray), A59H (blue), or K61H (green) mutants. Right: averaged Δratio. ($n = 20$). $P = 1.271 \times 10^{-6}$ for N264H. (One-way ANOVA with Dunnet's test, ***$P < 0.001$). **b** Mutated coordinating amino acid residues within 15 Å from N264 in the crystal structure of the VFT domain (PDB 1EWK). **c** Averaged Δratio induced by 10 μM Pd(bpy) (left, $P = 2.036 \times 10^{-6}$, $5.532 \times 10^{-6}$ for E60A/N264H, H111F/N264H, respectively) and 10 μM glutamate (right, $P = 1.349 \times 10^{-4}$ for H54F/N264H). ($n = 20$). (One-way ANOVA with Dunnet's test, ***$P < 0.001$). Data are presented as mean ± s.e.m.

**Design of low cytotoxic Pd(bpy) derivatives for neurons.** Palladium complexes have attracted much attention as anti-cancer drugs because their geometry and complex forming processes are similar to those of platinum-based drugs such as cisplatin[18,19]. The putative molecular mechanism of the anti-tumor activity underlies DNA binding of these metal complexes. However, neurotoxicity has been reported for the clinical use of platinum drugs[20]. Although previous reports show palladium complexes having favorable neurotoxicity when compared with that of platinum drugs, the neurotoxicity of palladium complexes needs to be examined when used in neurons or brain tissues.

Pd(bpy) toxicity was examined in PC12 cells, a neuron-like cell-line, or in primary cortical neurons (Fig. 4). The effect of Pd(bpy) on neuronal outgrowth, a characteristic feature of neurons, was evaluated in PC12 cells after differentiation of the cells by nerve growth factor (NGF). As shown in Fig. 4b, neurite outgrowth was not affected even at a high concentration of Pd(bpy) ([Pd(bpy) = 100 μM]). However, 30 μM Pd(bpy) clearly suppressed cell growth in undifferentiated PC12 cells (Fig. 4c). In primary cortical neurons, gradual $[Ca^{2+}]_i$ elevation was observed by treatment with 10 μM Pd(bpy), which became more prominent by 30 μM Pd(bpy) (Fig. 4d). Importantly, this $[Ca^{2+}]_i$ elevation was suppressed by neither the mGlu1 inhibitors (LY367385 and FITM) nor the PLC inhibitor (U73122) (Supplementary Fig. 9), which indicated a side effect of Pd(bpy) at the concentration used with the neurons. Considering the action of palladium complexes as anti-cancer drugs in the intracellular area, these unfavorable effects of Pd(bpy) are likely caused by cell permeability of the complex.

We then newly designed palladium complexes bearing hydrophilic or anionic ligands to suppress permeation of these compounds into cells (Fig. 4a). As shown in Fig. 4e, 3 μM Pd(OH-bpy) and Pd(sulfo-bpy) successfully activated mGlu1(N264H). However, Pd(EG-bpy) showed weakened responses, and Pd(di-sulfo-bpy) failed to activate mGlu1(N264H) at the same

concentration (Fig. 4e). The $EC_{50}$ values of Pd(OH-bpy) and Pd(sulfo-bpy) for mGlu1(N264H) activation were 1.1 and 0.62 μM, respectively (Fig. 4f). We next examined the effect of Pd(OH-bpy) or Pd(sulfo-bpy) on cell growth of PC12 cells. As with the case of Pd(bpy), cell growth was suppressed by 30 μM Pd(OH-bpy). In contrast, for Pd(sulfo-bpy), cell growth was unaffected even at the high concentration ([Pd(sulfo-bpy)] = 100 μM) (Fig. 4c). More critically, in cortical neurons, the influence on $[Ca^{2+}]_i$ elevation was not observed following treatment with 30 μM Pd(sulfo-bpy) (Fig. 4d), suggesting that the side effects were successfully suppressed using Pd(sulfo-bpy) bearing the anionic ligand. Consistently, the solubility of Pd(sulfo-bpy) was clearly improved when compared with that of Pd(bpy) (Fig. 4g). In the case of Pd(bpy), a linear increase of the absorbance at 312 nm was lost over 300 μM Pd(bpy) in artificial cerebrospinal fluid (ACSF), a buffer solution used for brain tissue experiments in the next section. In contrast, linearity of the absorbance was observed over 1 mM for Pd(sulfo-bpy), indicating that Pd(sulfo-bpy) stock solutions can be prepared in ACSF, which is another improvement when applying the palladium complex to brain slices.

**Knock-in of the N264H mutation does not alter mGlu1 function in mice.** With Pd(sulfo-bpy) showing low cytotoxicity in hand, we aimed to evaluate chemogenetic activation of endogenous mGlu1 to understand the physiological roles in the mouse brain. The mGlu family is composed of eight subtypes (mGlu1–8) divided into three groups (group I–III) according to their pharmacological properties[13]. mGlu1 belongs to the group I family with the cognate mGlu5 and is highly expressed in the dendritic regions of Purkinje cells in the cerebellum (Fig. 5a). The mGlu1 agonist 3,5-dihydroxyphenylglycine (DHPG) is typically used to examine the physiological role of mGlu1; however, DHPG cannot discriminate between mGlu1 and mGlu5. Moreover, mGlu5 is

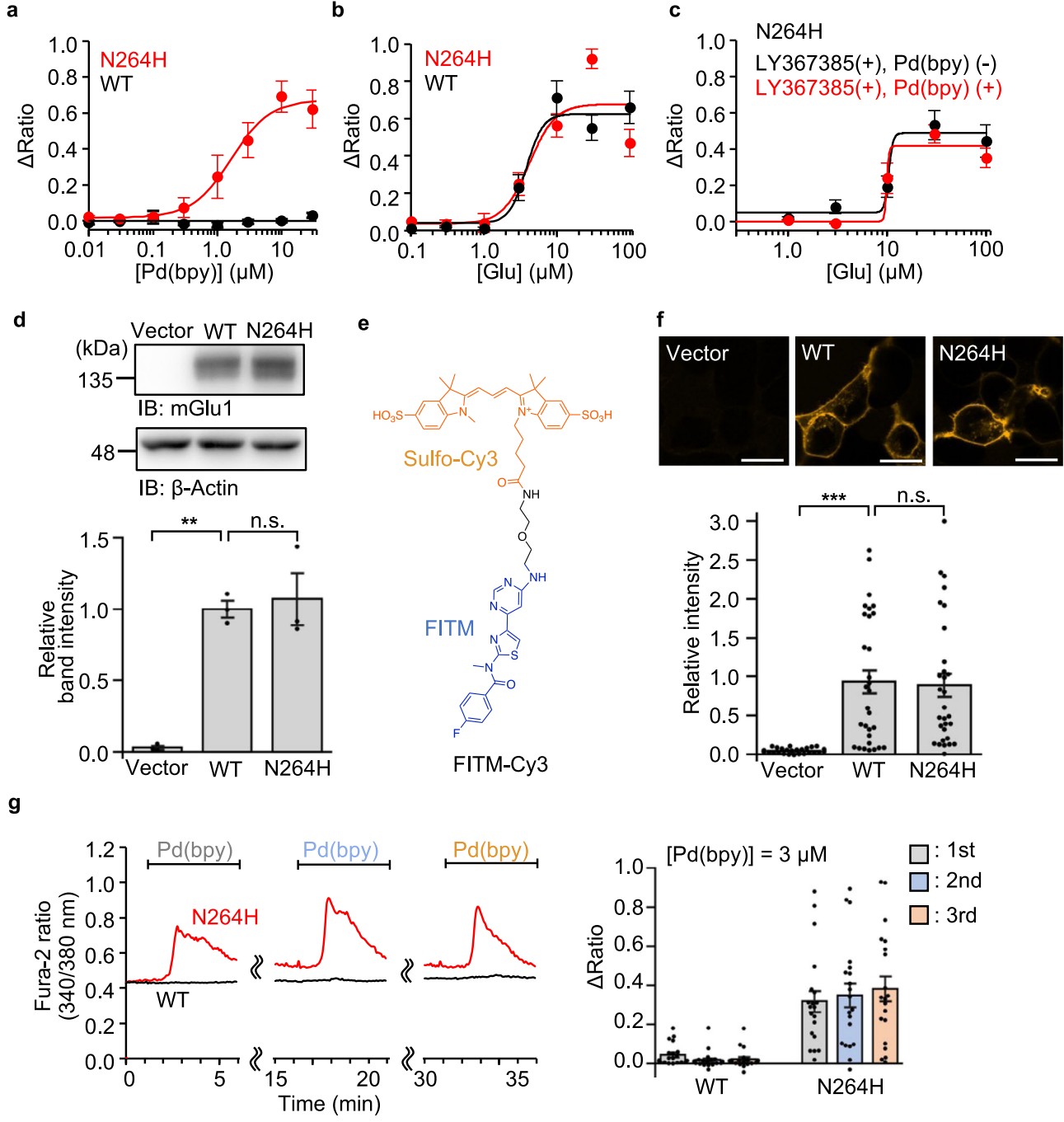

**Fig. 3 Characterization of mGlu1(N264H) mutant. a, b** Concentration-dependent curves for Pd(bpy) (**a**) and glutamate (**b**) in HEK293 cells expressing mGlu1(N264H) or WT mGlu1. ($n = 20$). **c** Evaluation of the positive allosteric effect of Pd(bpy) to the mGlu1(N264H) mutant. Effect of 3 μM Pd(bpy) on the concentration-dependency of the glutamate-responses in the presence of 10 μM LY367385 in HEK293 cells expressing mGlu1(N264H) is shown. ($n = 20$). **d** Evaluation of the protein expression level of mGlu1(N264H) by western blotting. ($n = 3$, biologically independent experiments, $P = 0.0015$, 0.8708 for WT, N264H, respectively). (One-way ANOVA with Dunnet's test, **$P < 0.01$, n.s. not significant). Full scan images are shown in Source Data. **e** Chemical structure of FITM-Cy3, a fluorescent probe for mGlu1. **f** Evaluation of the cell-surface expression of mGlu1(N264H) by live imaging using FITM-Cy3. Upper: Representative confocal images of the HEK293 cells transfected with WT mGlu1, mGlu1(N264H), or vector control after addition of 1 μM FITM-Cy3. Lower: quantification of fluorescent intensity on the cell surface. ($n = 30$, $P = 0.0008$, 0.9926 for WT, N264H, respectively). Scale bars, 20 μm. (One-way ANOVA with Dunnet's test, ***$P < 0.001$, n.s. not significant). **g** The $Ca^{2+}$ response induced by the repetitive treatment of 3 μM Pd(bpy). Left: representative trace of the $Ca^{2+}$ response induced by 3 μM Pd(bpy) in HEK293 cells expressing WT mGlu1 (black) or mGlu1(N264H) (red). Pd(bpy) was treated repetitively after washing the cells with the culture medium and measurement buffer for 10 min. Right: averaged Δratio induced by 3 μM Pd(bpy). ($n = 20$). Data are presented as mean ± s.e.m.

expressed in neurons and non-neuronal (glial) cells such as astrocytes[21]. Thus, the contribution of DHPG in brain tissues is complicated. Alternatively, mGlu1 selective PAM reagents would be useful; however, the effect of PAM reflects the difference in the extracellular glutamate concentration, as described in the Introduction. Therefore, a direct activation method of mGlu1, such as our dA-CBC strategy, should be ideal to understand the physiological roles of mGlu1 in brain tissues.

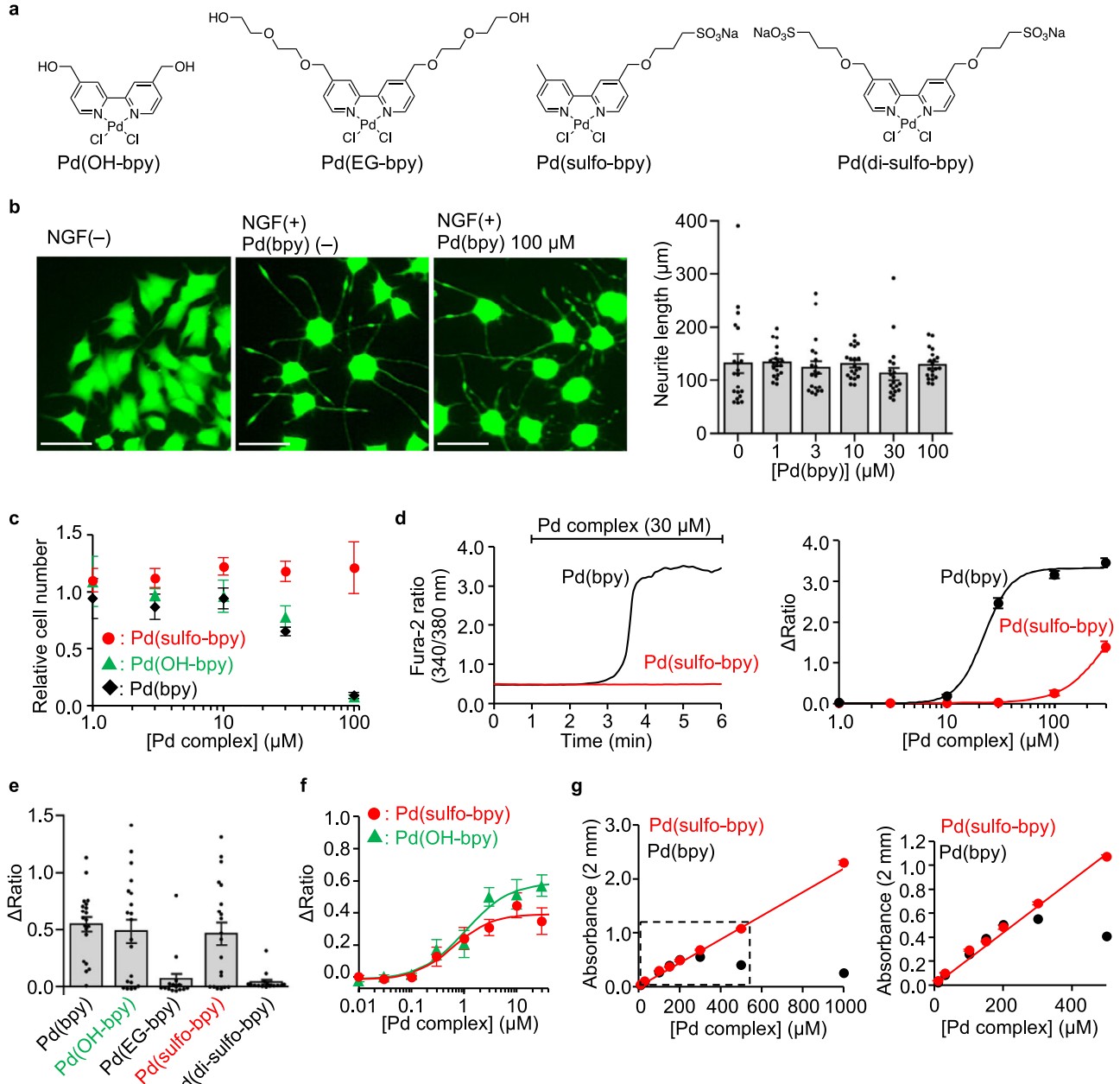

**Fig. 4 Low toxic Pd(bpy) derivatives for neuronal studies. a** Chemical structures of the Pd(bpy) derivatives. **b** Evaluation of the effect of Pd(bpy) for NGF-induced neurite outgrowth of PC12 cells. Left: representative fluorescent images of PC12 cells treated with or without Pd(bpy). PC12 cells were visualized using Calcein-AM. Right: concentration-dependency of Pd(bpy) for neurite outgrowth in the PC12 cells after treatment of 1 ng/mL NGF. ($n = 20$). Scale bars, 50 μm. **c** Concentration-dependency of Pd(bpy) derivatives on the cell growth in undifferentiated PC12 cells. ($n = 3$). **d** The abnormal $Ca^{2+}$ response by Pd(bpy) (black) or Pd(sulfo-bpy) (red) in the cultured cortical neurons. Left: representative trace of the $Ca^{2+}$ response induced by 30 μM Pd(bpy) (black) or Pd(sulfo-bpy) (red). Right: concentration-dependence curves for Pd(bpy) (black) or Pd(sulfo-bpy) (red). ($n = 30$). **e** $Ca^{2+}$ responses of Pd(bpy) derivatives for mGlu1(N264H) mutant in HEK293 cells. Left: averaged Δratio induced by 3 μM Pd(bpy) derivatives. ($n = 15$ for Pd(EG-bpy) and Pd(di-sulfo-bpy). $n = 20$ for Pd(bpy), Pd(OH-bpy), and Pd(sulfo-bpy)). Data are presented as mean ± s.e.m. **f** Concentration-dependent curves for Pd(sulfo-bpy) (red) and Pd(OH-bpy) (green) in HEK293 cells expressing the mGlu1(N264H) mutant ($n = 20$). **g** Solubility assay of the Pd(bpy) (black) and Pd(sulfo-bpy) (red) in ACSF using UV-vis spectroscopy. Linear increment of absorbance at 312 nm corresponding to the palladium complexes indicates the solubility of the complexes. The concentration dependency of the absorbance of Pd complexes in the high or low concentration range is shown in the left or right, respectively ($n = 3$). Data are presented as mean ± s.e.m.

We generated knock-in (KI) mice harboring the point mutation (N264H) in the mouse gene using the CRISPR/Cas9 system[22] to chemogenetically control endogenous mGlu1 function (Fig. 5b and Supplementary Fig. 10a). The KI mouse was termed the coordination-based chemogenetics (CBC) ready mouse (*mGlu1CBC/CBC* and *mGlu1CBC/+* for homozygous and heterozygous, respectively). Bodyweight and gross anatomy of the cerebellum were unaffected in *mGlu1CBC/CBC* mice (Supplementary Fig. 10b, c). However, given the severe ataxic phenotypes in *mGlu1* knock-out (*mGlu1−/−*) mice[23,24] and autosomal-recessive congenital cerebellar ataxia by familial mutations in the *mGlu1* gene in humans[25–27], we analyzed *mGlu1CBC/CBC* mice in detail.

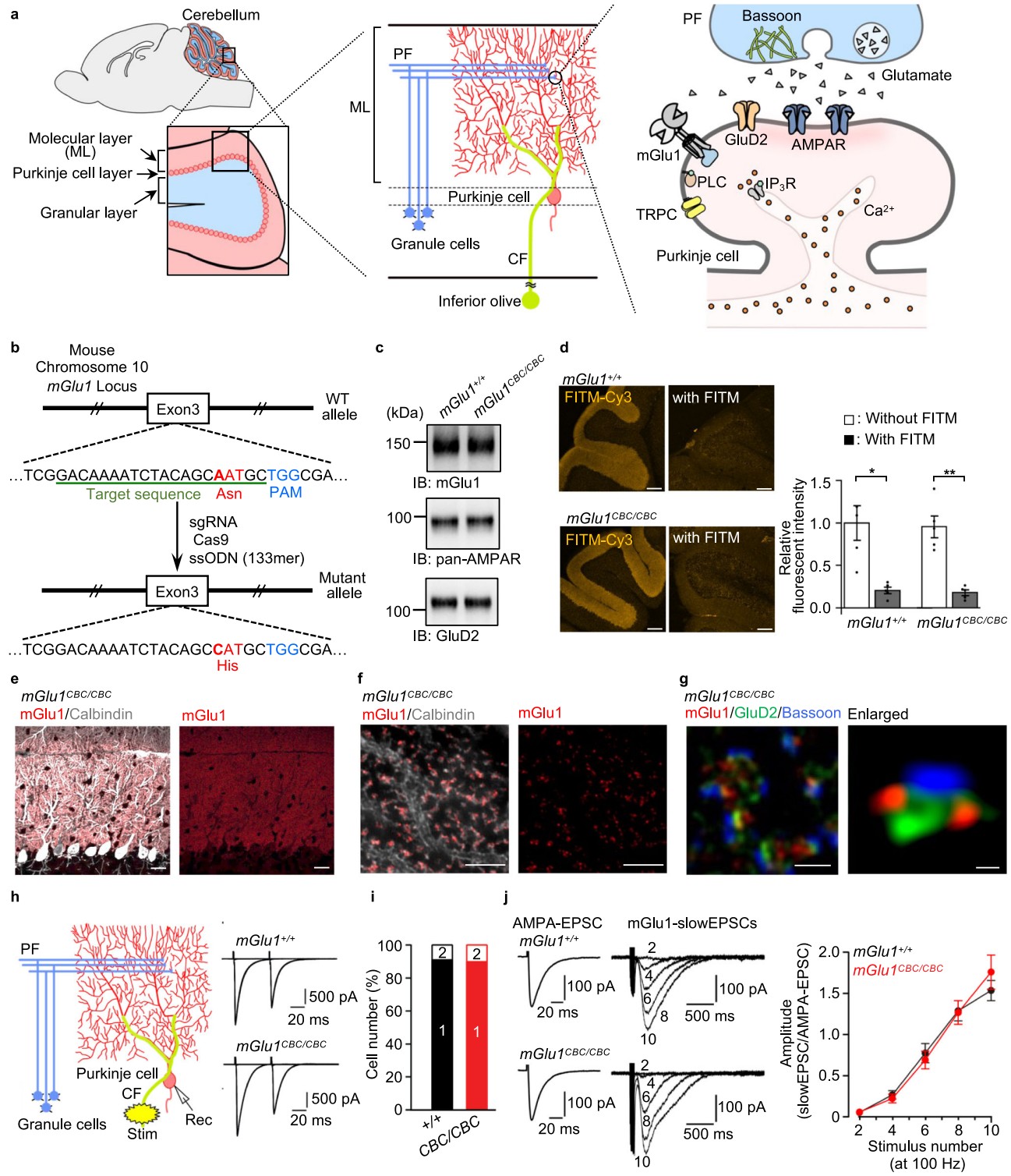

Western blotting of mGlu1 indicated that the expression level of mGlu1 was unchanged in the cerebellum of $mGlu1^{CBC/CBC}$ mice (Fig. 5c). In addition, the expression of excitatory postsynaptic glutamate receptors [AMPAR and δ2-type glutamate receptor (GluD2)] were also unaffected in the mutant mice. The cell-surface expression level of mGlu1 in the acutely prepared cerebellar slices was examined using FITM-Cy3. Prominent fluorescence was observed from the molecular layer (ML) of the cerebellum, where mGlu1 is highly expressed (Fig. 5a, d and Supplementary Fig. 10d). Given the membrane impermeability of

FITM-Cy3 and disappearance of fluorescence in the co-presence of excess FITM, the fluorescent signals correspond to cell-surface mGlu1 and surface mGlu1 levels were comparable between WT ($mGlu1^{+/+}$) and $mGlu1^{CBC/CBC}$ mice.

Next, we evaluated the distribution of the mGlu1(N264H) mutant in the cerebellum. Immunohistochemistry using an anti-mGlu1 antibody clearly indicated that mGlu1 localized in the dendritic spines of Purkinje cells (Fig. 5e, f and Supplementary Fig. 11a, b). Notably, super-resolution microscopy showed that immune-positive signals for mGlu1 were predominantly localized

**Fig. 5 Chemogenetic regulation of endogenous mGlu1 in brain tissue. a** Illustration of a sagittal section of the mouse brain and the enlarged image of the cerebellum (left), major components of the cerebellar cortical neuronal circuits (middle), and PF−Purkinje cell synapses (right). IP$_3$R, inositol triphosphate receptor; TRPC, transient receptor potential channel canonical. **b** Illustration for preparing mGlu1(N264H) KI mice (*mGlu1$^{CBC/CBC}$*) using CRISPR-Cas9 system. **c** Evaluation of the protein expression level of mGlu1, AMPAR and GluD2 in the cerebellum between *mGlu1$^{+/+}$* and *mGlu1$^{CBC/CBC}$* by western blotting. Full scan images are shown in Source Data. **d** Quantification of the surface expression level of mGlu1 in acute cerebellar slices between *mGlu1$^{+/+}$* and *mGlu1$^{CBC/CBC}$* using FITM-Cy3. Scale bars, 200 μm. (*n* = 5, *P* = 0.01835, 0.001975 for *mGlu1$^{+/+}$*, *mGlu1$^{CBC/CBC}$*, respectively). (Two-tailed Welch's *t*-test **P* < 0.01, *\*P* < 0.05). See Supplementary Figure 10d for the merge images with Hoechst33342. **e, f** Conventional confocal (**e**) and super-resolution (**f**) microscopic images of immune-positive signals for mGlu1 (red) and calbindin (white) in the ML of *mGlu1$^{CBC/CBC}$* cerebellar slices. Scale bars, 20 μm (**e**) and 2 μm (**f**). **g** Super-resolution microscopic images of immune-positive signals for mGlu1 (red), GluD2 (green), and Bassoon (blue). Scale bar, 1 μm. An enlarged view is shown in a right panel. Scale bar, 200 nm. **h** Representative CF-EPSC traces from *mGlu1$^{+/+}$* (top) and *mGlu1$^{CBC/CBC}$* (bottom). In the left, an orientation of stimulus and recording electrodes is shown. **i** A histogram showing the percentage of the number of CFs innervating single Purkinje cells. [*n* = 32 cells from 5 mice (*mGlu1$^{+/+}$*) or 30 cells from 5 mice (*mGlu1$^{CBC/CBC}$*)]. **j** mGlu1-mediated slowEPSC data. By adjusting PF stimulus intensities, similar sizes of PF-evoked AMPA-EPSC (left traces) were obtained. Then, slowEPSC (right traces) were recorded by application of 2−10 times of PF stimuli at 100 Hz in the presence of AMPAR blockers. Amplitudes of slowEPSC were normalized by amplitudes of AMPA-EPSCs and plotted against the number of stimulations (right graph). [*n* = 9 cells from 3 mice (*mGlu1$^{+/+}$*) or 13 cells from 4 mice (*mGlu1$^{CBC/CBC}$*)]. Data are presented as mean ± s.e.m.

at the edges of synaptic contacts, which were represented by signals for a parallel fiber (PF)–Purkinje cell synapse specific-postsynaptic protein (GluD2) and a presynaptic protein (bassoon), indicating perisynaptic localization of mGlu1 in both *mGlu1$^{+/+}$* and mutant *mGlu1$^{CBC/CBC}$* mice (Fig. 5g and Supplementary Fig. 11c). This result is consistent with the similar subcellular localization of mGlu1 revealed by immunoelectron microscopy[27].

Subsequently, the function of the mGlu1(N264H) mutant was evaluated by whole-cell patch-clamp recordings of Purkinje cells in the cerebellar slices. We first confirmed that excitatory postsynaptic currents (EPSCs) evoked by PF and climbing fiber (CF) stimuli (PF-EPSCs and CF-EPSCs, respectively) were normal in amplitude and the paired-pulse ratio, reflecting presynaptic functions[28], of *mGlu1$^{CBC/CBC}$* mice (Fig. 5h and Supplementary Fig. 12). Previous studies have revealed that mGlu1 is required for synapse elimination of surplus CF inputs to Purkinje cells during development to make a one-to-one relationship in adults, and the relationship is impaired in *mGlu1*-deficient mice[27]. In this context, 90% of Purkinje cells had a single step of CF-EPSC amplitude by increasing CF stimuli in adult *mGlu1$^{CBC/CBC}$* mice, as with the case of *mGlu1$^{+/+}$* (91%, *P* = 0.934 by Mann–Whitney *U* test; Fig. 5i), suggesting that mGlu1(N264H) is functioning normally for CF synapse elimination during development. Furthermore, burst PF stimulation (2–10 times of PF stimuli at 100 Hz) induced robust TRPC- or TRPC/GluD2-mediated slow currents[29,30] designated as slowEPSCs by activation of the perisynaptic mGlu1 in both *mGlu1$^{CBC/CBC}$* and *mGlu1$^{+/+}$* Purkinje cells (Fig. 5j, *P* = 0.898 by two-way repeated ANOVA). Because these slowEPSCs are often used to evaluate the degree of mGlu1 activation[31], these results indicate that the mGlu1(N264H) mutant functions in a neuronal activity-dependent manner.

Consistent with the intact subcellular localization and function of the mGlu1(N264H) mutant endogenously expressed in mice, the ataxic phenotype observed in *mGlu1$^{−/−}$* mice was never observed in the *mGlu1$^{CBC/CBC}$* mice (Supplementary Movie 1). In addition, *mGlu1$^{CBC/CBC}$* mice showed high performance in the rota-rod test (Supplementary Fig. 13), which contrasts to *mGlu1$^{−/−}$* mice whose behaviors were severely impaired in motor coordination, as reported previously[23,24]. Taken together, the introduction of the N264H mutation in mGlu1 does not affect original receptor activity in vivo.

**Induction of synaptic plasticity by mGlu1 dA-CBC in the cerebellum.** In brains, synaptic plasticity such as long-term potentiation (LTP) and LTD of excitatory synaptic transmission has been proposed as a cellular substrate for learning and memory[32,33]. In particular, cerebellar LTD, which occurs at

PF–Purkinje cell synapses by the simultaneous activation of PF and CF inputs, generally underlies a certain form of motor-related learning. mGlu1 is highly expressed at PF–Purkinje cell synapses as described (Fig. 5a, d) and absolutely required for the induction of cerebellar LTD. We first examined whether cerebellar LTD occurs in acute cerebellar slices from *mGlu1$^{CBC/CBC}$* mice following electric stimulation (Fig. 6a). When PF stimuli were applied repetitively and simultaneously with Purkinje cell depolarization, mimicking a CF input, at 1 Hz (LTD-stim), a robust LTD occurred in *mGlu1$^{CBC/CBC}$* mice (68 ± 4% at *t* = 26−30 min) and *mGlu1$^{+/+}$* mice (72 ± 5% at *t* = 26−30 min, *P* = 0.628 by Mann–Whitney *U* test; Fig. 6b). Furthermore, application of the mGlu1/mGlu5 activator, DHPG (50 μM for 10 min under a current-clamp mode) also caused a chemical-induced LTD (chemLTD) in cerebellar slices from *mGlu1$^{+/+}$* (55 ± 7% at *t* = 31−35 min) and *mGlu1$^{CBC/CBC}$* (56 ± 6% at *t* = 31−35 min, *P* = 1.000 by Mann–Whitney *U* test; Fig. 6c) mice. These results also emphasize that the mGlu1(N264H) mutant works functionally.

We then examined the direct activation of mGlu1 using dA-CBC in brain slices to clarify the contribution of mGlu1 activation in LTD. For this purpose, 1 μM Pd(sulfo-bpy) was used as the treatment instead of 50 μM DHPG. As shown in Fig. 6d, after washing out Pd(sulfo-bpy), sustained decreases of the PF-EPSC were observed in *mGlu1$^{CBC/CBC}$* mice (71 ± 5% at *t* = 31−35 min). In contrast, the decrease was not observed by treating *mGlu1$^{+/+}$* mice with Pd(sulfo-bpy) (99 ± 4% at *t* = 31−35 min, *P* = 0.011 vs *mGlu1$^{CBC/CBC}$* mice by Kruskal-Wallis test followed by the Scheffe post hoc test), indicating successful activation of endogenous mGlu1 and induction of synaptic plasticity using Pd(sulfo-bpy). Interestingly, Pd(sulfo-bpy)-induced chemLTD was also observed in heterozygous *mGlu1$^{CBC/+}$* mice (68 ± 5% at *t* = 31−35 min, *P* = 0.009 vs *mGlu1$^{+/+}$* mice by Kruskal–Wallis test followed by the Scheffe post hoc test; Fig. 6d) and similar to homozygous *mGlu1$^{CBC/CBC}$* mice (*P* = 0.964 vs *mGlu1$^{CBC/CBC}$* by Kruskal–Wallis test followed by the Scheffe post hoc test; Fig. 6d), suggesting that a half amount of mGlu1 in *mGlu1$^{+/+}$* mice is sufficient to induce chemLTD. Collectively, dA-CBC was successfully applied for chemogenetic activation of mGlu1 in mice, which clarified the essential roles of mGlu1 activation for LTD, the putative mechanism of motor learning in the cerebellum.

**Cell-type-specific activation of mGlu1 in the cerebellum using dA-CBC.** mGlu1 is widely expressed not only in the cerebellum but also in various brain regions such as the olfactory bulb, thalamus, and hippocampus[24]. Therefore, to understand the roles of mGlu1 in each brain region, a cell- or region-type-specific

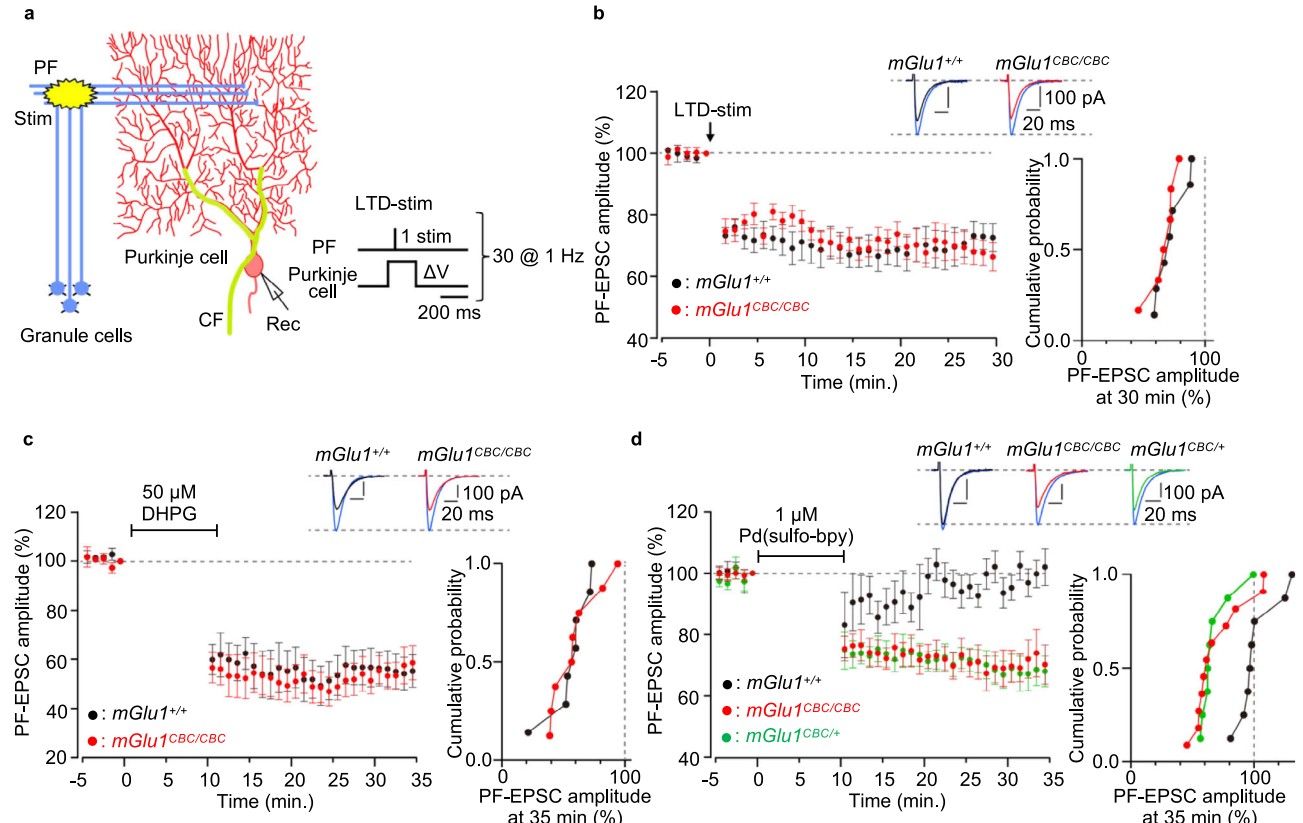

**Fig. 6 Cerebellar LTD is induced by direct activation of the mGlu1 mutant by dA-CBC. a** A schematic diagram illustrating the experimental set-up. Conjunctive stimulus [PF/ΔV-stim, 30× (PF stimuli plus Purkinje cell depolarization) at 1 Hz] was applied to induce LTD at PF–Purkinje cell synapses (LTD-stim). An orientation of stimulus and recording electrodes and stimulus conditions are shown. **b** Averaged data of LTD recordings from *mGlu1*[+/+] (black circles) and *mGlu1*[CBC/CBC] (red circles). Insets, representative PF-EPSC traces just before (blue; *t* = −1 min) and 30 min after (black and red) LTD-stim in *mGlu1*[+/+] and *mGlu1*[CBC/CBC] mice, respectively. A right graph shows a cumulative probability of the degree of LTD at *t* = 30 min. [*n* = 7 cells from 4 mice (*mGlu1*[+/+]) or 6 cells from 4 mice (*mGlu1*[CBC/CBC])]. **c, d** Averaged data of DHPG-induced (**c**) or Pd(sulfo-bpy)-induced (**d**) chemLTD recordings from *mGlu1*[+/+] (black circles), *mGlu1*[CBC/CBC] (red circles) and *mGlu1*[CBC/+] (green circles in **d**) mice. Insets, representative PF-EPSC traces just before (blue; *t* = −1 min) and 25 min after (black, red, or green) drug washout in each mouse group. A right graph shows a cumulative probability of the degree of LTD at *t* = 35 min. [*n* = 7 cells from 4 mice (*mGlu1*[+/+]) or 8 cells from 5 mice (*mGlu1*[CBC/CBC]) in **c** and *n* = 8 cells from 5 mice (*mGlu1*[+/+]), 11 cells from 6 mice (*mGlu1*[CBC/CBC]) or 8 cells from 5 mice (*mGlu1*[CBC/+]) in **d**]. Data are presented as mean ± s.e.m.

mGlu1 activation system is desirable. For this purpose, we examined the applicability of the dA-CBC strategy for cell-type-specific activation of mGlu1 in cerebellar slices using adeno-associated viruses (AAVs) with each cell-type-specific promoter [L7 or human synapsin (hSyn) for Purkinje cells or ML inter-neurons (MLIs), respectively]. Considering the strict packaging capacity limit (5.2 kb) of the AAV and the large transgene size (3.6 kb) of mGlu1 cDNA, we applied two-component AAVs using the tetracycline (tet)-inducible system — a tet-off transac-tivator (tTA) and a tet-responsive element (TRE)[34] (Fig. 7a and Supplementary Fig. 14a). In this system, Purkinje cell-specific expression of tTA should be achieved using an AAV vector (pAAV-L7-GFP-IRES-tTA) with the L7 promoter (0.8 kb), in which GFP is co-expressed under the same promoter as a infection marker. A second AAV vector encodes mGlu1 under a short TRE promoter (0.3 kb) [pAAV-TRE-HA-mGlu1 WT or pAAV-TRE-HA-mGlu1(N264H)], in which a short regulatory element [WPRE3 and SV40 polyA (0.4 kb)][35] is used instead of the conventional regulatory element [WPRE and bGH polyA (1.1 kb)]. We expected that these shorter elements allow accep-tance of the large mGlu1 cDNA even in the packaging capacity limit of AAVs.

For selective expression of mGlu1(N264H) in Purkinje cells, a mixture of AAV-TRE-HA-mGlu1(N264H) and AAV-L7-GFP-

IRES-tTA, termed AAV-L7-mGlu1[CBC], was injected directly into the cerebellum of WT mice, and brains were fixed and sliced 8 days after viral infection. After immunostaining the slices, the immunofluorescent signals of GFP merged well with the anti-carbonic anhydrase VIII (Car8) signals, a Purkinje cell-specific marker (Fig. 7b). Immunostaining using an anti-HA antibody indicated that HA-tagged mGlu1(N264H) was detected in GFP-positive cells. Consistently, western blotting using the anti-HA antibody revealed that the molecular weight of the transgene product corresponds to that of mGlu1 (Fig. 7c). In western blotting analyses, the anti-mGlu1 signals in the AAV-infected mice were stronger than those in non-infected mice. These results indicate that mGlu1(N264H) was abundantly expressed in a Purkinje cells-specific manner using the AAVs. In addition, similar results were obtained for mice infected with a mixture of AAV-TRE-mGlu1(WT) and AAV-L7-GFP-IRES-tTA, termed AAV-L7-mGlu1[WT] (Supplementary Fig. 14b).

To examine whether the AAV-mediated dA-CBC strategy works, we next recorded chemLTD using acutely prepared cerebellar slices from WT mice infected with AAV-L7-mGlu1[CBC]. As shown in Fig. 7d, bath application of 1 μM Pd(sulfo-bpy) prominently induced chemLTD in the GFP-positive Purkinje cells (66 ± 5% at *t* = 31−35 min), which is consistent with the results using *mGlu1*[CBC/CBC] or *mGlu1*[CBC/+]

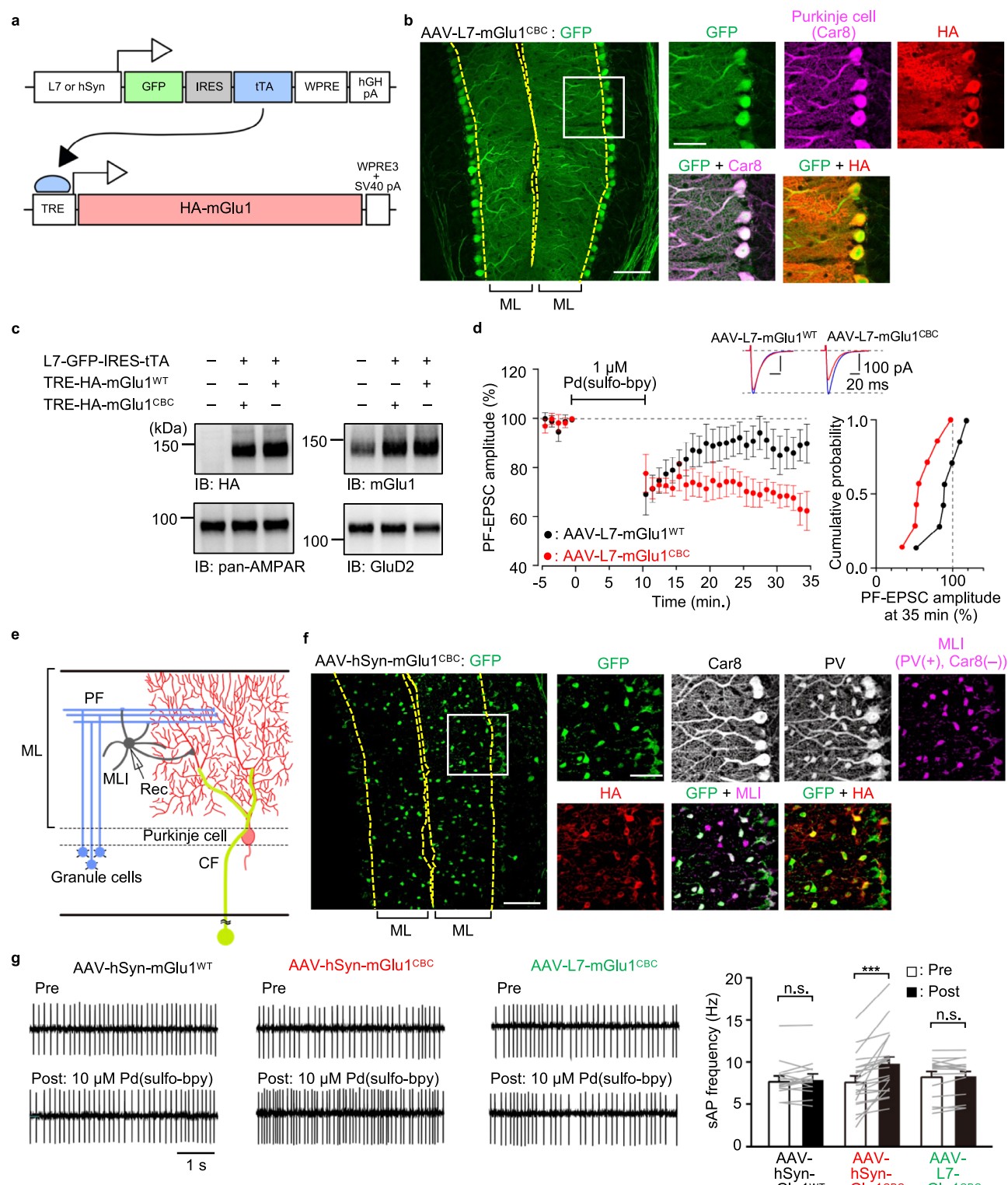

mice (Fig. 6d). In contrast, Pd(sulfo-bpy)-mediated chemLTD was not observed in the GFP-positive Purkinje cells of the slices infected with AAV-L7-mGlu1^WT ($88 \pm 7\%$ at $t = 31-35$ min, $P = 0.047$ by Mann–Whitney $U$ test; Fig. 7d). These observations indicate that the AAV-mediated dA-CBC strategy works well for cell-type-specific activation of mGlu1 in cerebellar slices.

Finally, to establish the potential utility of the AAV-mediated dA-CBC strategy, mGlu1(N264H) was expressed in MLIs of the cerebellum, where mGlu1 is highly expressed[36,37] (Fig. 7e), using the AAV vector with the hSyn promoter. Eight days after co-

injection of AAV-TRE-mGlu1(N264H) with AAV-hSyn-GFP-IRES-tTA, termed AAV-hSyn-mGlu1^CBC, into the cerebellum of WT mice, the immunofluorescent signals of GFP were detected primarily from MLIs (Car8-negative and parvalbumin (PV)-positive cells) in the cerebellar slice (Fig. 7f). The GFP signals merged well with the anti-HA signals, indicating the selective expression of mGlu1(N264H) in MLIs (Fig. 7f and Supplementary Fig. 14c). Importantly, in MLIs, several groups have reported that activation of mGlu1 increases the excitability in MLIs, which was revealed by bath application of the mGlu1 activator,

**Fig. 7 Cell-type-specific activation of mGlu1 in the cerebellum using dA-CBC and AAVs. a** Schematic illustration of the two-component AAV system for Purkinje cells- or MLIs-specific expression of mGlu1. **b** Confocal microscopic images of immune-positive signals for GFP, Car8 and HA-tag in the cerebellar slice infected with AAV-L7-mGlu1$^{CBC}$. The white squarer ROI is expanded on the right. Scale bars, 100 μm (in a large left panel) and 50 μm (in small right panels). **c** Evaluation of expression level of HA-tagged mGlu1, AMPAR and GluD2 in the AAV-infected or -uninfected cerebellum by western blotting. Full scan images are shown in Source Data. **d** Averaged data of Pd(sulfo-bpy)-induced chemLTD recordings from AAV-L7-mGlu1$^{WT}$ (black circles) and AAV-L7-mGlu1$^{CBC}$ (red circles). Insets, representative PF-EPSC traces just before (blue; $t = -1$ min) and 25 min after (black or red) drug washout in each mouse group. A right graph shows a cumulative probability of the degree of LTD at $t = 35$ min. [$n = 7$ cells from 5 mice (AAV-L7-mGlu1$^{WT}$) or 7 cells from 5 mice (AAV-L7-mGlu1$^{CBC}$)]. **e** Schematic illustration of major components of the cerebellar cortical neuronal circuits including MLI. An orientation of recording electrode is shown. **f** Confocal microscopic images of immune-positive signals for GFP, Car8, PV and HA-tag in the cerebellar slice infected with AAV-hSyn-mGlu1$^{CBC}$. The white squarer ROI is expanded on the right. To obtain the MLI image, immunofluorescent signals in the Car8 image were subtracted from those in the PV image. Scale bars, 100 μm (in a large left panel) and 50 μm (in small right panels). **g** Spontaneous action potential (sAP)-mediated currents from MLIs infected with AAV-hSyn-mGlu1$^{WT}$, AAV-hSyn-mGlu1$^{CBC}$, or AAV-L7-mGlu1$^{CBC}$ with bath application of 10 μM Pd(sulfo-bpy). Upper: sAP-mediated currents before Pd(sulfo-bpy) treatment. Lower: sAP-mediated currents after Pd(sulfo-bpy) treatment. [$n = 16$ cells from 4 mice (AAV-hSyn-mGlu1$^{WT}$, $P = 0.4534$), $n = 22$ cells from 4 mice (AAV-hSyn-mGlu1$^{CBC}$, $P = 4.613 \times 10^{-5}$) or 15 cells from 4 mice (AAV-L7-mGlu1$^{CBC}$, $P = 0.2934$)]. (two-sided Wilcoxon signed-rank test, ***$P < 0.001$, n.s. not significant). Data are presented as mean ± s.e.m.

DHPG[36,37]. However, it is not completely clear whether this effect is mediated by the direct activation of mGlu1 expressed in MLIs since $[Ca^{2+}]_i$ increases in Purkinje cells could retrogradely activate MLIs by the dendritic release of glutamate[38]. To clarify this point, we examined selective activation of mGlu1(N264H) in MLIs using our system. Consistent with previous reports[36,37], bath application of DHPG clearly increased the number of spontaneous action potentials (sAP) in both AAV-hSyn-mGlu1$^{CBC}$ and AAV-hSyn-mGlu1$^{WT}$-infected MLIs (Supplementary Fig. 14d). Notably, as shown in Fig. 7g, 10 μM of Pd(sulfo-bpy) increased the firing rate of MLIs infected with AAV-hSyn-mGlu1$^{CBC}$ effectively ($P = 4.613 \times 10^{-5}$ by Wilcoxon signed-rank test) but not MLIs infected with AAV-hSyn-mGlu1$^{WT}$ ($P = 0.453$ by Wilcoxon signed-rank test). In contrast, Pd(sulfo-bpy) failed to increase the firing rate of MLIs surrounded by mGlu1(N264H)-expressing Purkinje cells that were infected with AAV-L7-mGlu1$^{CBC}$ ($P = 0.293$ by Wilcoxon signed-rank test; Fig. 7g). These results indicate that mGlu1 in MLIs, but not in Purkinje cells, directly regulates the excitability of MLIs. Taken together, cell-type-specific activation of mGlu1 was demonstrated successfully using the dA-CBC strategy and AAV techniques.

## Discussion
We developed a direct-activation method of mGlu1 without affecting glutamate affinity using dA-CBC, in which a single point mutation (N264H) was sufficient for chemogenetics activation by Pd(bpy). To demonstrate the chemogenetic activation of mGlu1 endogenously expressed in neurons, we generated mGlu1(N264H)-KI ($mGlu1^{CBC/CBC}$) mice using the CRISPR-Cas9 system and obtained $mGlu1^{CBC/CBC}$ mice that exhibited no obvious abnormal phenotypes. Pd(sulfo-bpy), which showed low toxicity to neurons, successfully activated the mGlu1 mutant in the cerebellar slice prepared from $mGlu1^{CBC/CBC}$ mice. Notably, Pd(sulfo-bpy) induced chemLTD in cerebellar slices, which indicated that activation of endogenous mGlu1 is sufficient to evoke the cellular basis of motor learning. Moreover, dA-CBC was applied successfully for Purkinje cells- or MLI-specific activation of mGlu1 using AAVs, which shows the potential utility of this chemogenetic technique for understanding the physiological roles of mGlu1 in a cell-type-specific manner.

Genetic studies indicate that mGlu1 has essential roles in motor learning. Global mGlu1 knock-out (KO) mice show an ataxic phenotype and lack cerebellar LTD[23,24]. Introduction of a mGlu1 transgene with a Purkinje cell-specific L7 promoter into the mGlu1 KO mice restored the cerebellar LTD, suggesting essential roles of mGlu1 in cerebellar function[39]. In addition, mGlu1 conditional knock-out (cKO) using the tetracycline-controlled gene regulation

system revealed mGlu1 function in adult mice[40]. These genetic approaches are undoubtedly powerful in understanding the physiological roles of mGlu1. However, the expression level and transcription timing are different when compared with those of the native system. Moreover, in the gene knock-out or transgene approaches, adaptive compensation of related genes or fluctuation of the expression level of related proteins are issues. In fact, the expression level of mGlu5 in Purkinje cells is significantly increased in both mGlu1 cKO mice and spinocerebellar ataxia type 1 mice showing decreased expression of mGlu1[41,42]. In contrast, in the mGlu1 dA-CBC strategy, introducing a single point mutation (N264H) was sufficient for chemogenetic modulation, allowing preparation of knock-in ($mGlu1^{CBC/CBC}$) mice, in which the expression and function of the receptors were not affected. Indeed, we successfully controlled endogenous mGlu1 function using low cytotoxic Pd(sulfo-bpy) in acute cerebellar slices. Although in vivo application of this technique has not been examined, this approach has the potential for in vivo regulation of endogenous mGlu1 if the palladium complex can be delivered efficiently to brains.

In addition to well-used chemogenetic methods such as DREADD[6–8] and PSAM/PSEM[9,10], some potential approaches for selective activation of target receptors have been achieved by chemical biology manipulations[43,44]. For example, metal coordination has been used for functional switching of cell-surface receptors, in which the ligand-binding region of GPCRs has been mutated into the metal coordination site for metal-induced activation[45,46]. However, native ligand-binding is affected in most cases. As an alternative, tethering photo-switchable ligands to the target protein has been demonstrated successfully via cysteine modification[47,48] or a self-labeling enzyme tag[48,49], in which mutations were introduced outside or in regions distal from the ligand-binding site. Thus, native ligand-binding properties are not affected, which enabled photo-switching of particular types of neurotransmitter receptors, including glutamate receptors. Notably, photo-switching of endogenous ion-channel-type γ-aminobutyric acid A (GABA$_A$) receptors has been demonstrated using mice where the corresponding cysteine mutation was introduced into the genomic region encoding the GABA$_A$ receptor[50]. However, chemogenetic control of endogenous receptors is still very challenging. In this context, the current dA-CBC method, which enables control of endogenous mGlu1 function in cerebellar slices prepared from mGlu1(N264H)-knock-in mice, is a notable chemogenetic strategy for controlling endogenous receptor function.

Furthermore, a cell-type-specific dA-CBC method was established by combining this method with the AAV expression system. Using this strategy, we successfully activated mGlu1 expression in Purkinje cells or MLIs independently in the cerebellum. Although

mGlu1 in Purkinje cells has been studied extensively by analyzing Purkinje cell-specific mGlu1 transgenic rescue mice[39,40], physiological roles of mGlu1 in MLIs are largely unknown. Bath application of DHPG reportedly enhances cellular excitability of MLIs in acute cerebellar slices[36,37]; however, it remains unclear which cell (i.e., Purkinje cells or MLIs) or which type (i.e., mGlu1 or mGlu5) of mGlu is responsible for the event. By using a cell-type-specific dA-CBC method, we found that mGlu1 in MLIs directly regulates the excitability of neurons. Although mGlu-dependent excitation has been observed in different brain regions or different types of neurons, such as hippocampal CA1 pyramidal neurons, striatal dopaminergic neurons and retinal ganglion cells, the downstream signals that induce neuronal depolarization and inward currents are unique in each cell-type[13,51,52]. Among them, delta-type glutamate receptors (GluDs) are reported to open the gate following mGlu1 activation in Purkinje cells and striatal dopaminergic neurons[51]. Interestingly, GluDs are enriched in MLIs and Purkinje cells in the cerebellum[53]. Future efforts will use our cell-type-specific dA-CBC method to determine which downstream signals regulate MLI excitability. Thus, the cell-type-specific dA-CBC strategy is a powerful and general tool for mGlu1 activation in the brain.

The dA-CBC strategy for mGlu1 is based on structural changes to the VFT domain. Initial X-ray structural analyses of the VFT domain of mGlu1 with glutamate showed a conformational change from a resting open- to an active closed state[14,15]. However, both agonist-bound open and antagonist-bound closed states have also been reported in structural analyses. Regarding these controversial observations, FRET-based analyses[54] on live cells and recent structural analyses of full-length mGlu receptors using cryo-electron microscopy[55–57] strongly suggest that closing of the VFT domain correlates with receptor activation. As described, mGlu1 belongs to the class C GPCR family. The class C GPCRs include mGlu receptors, GABA$_B$ receptors, Ca$^{2+}$-sensing receptors and sweet and amino acid taste receptors, which share the large extracellular ligand-binding domain as a unique structural feature. Notably, structural analyses of the ligand-binding domain or full-length of these receptors have been reported recently[55–60]. These studies indicate that structural changes to the ligand-binding domain, known as the Venus Flytrap mechanism, are essential for activating class C GPCRs. Thus, our dA-CBC, a structure-based chemogenetic approach, can be potentially used to understand the physiological roles of other class C GPCR subtypes.

## Methods

All synthesis procedures and compound characterizations, inositol-1-phosphate (IP1) assay, gross cerebellar anatomy and rota-rod test are described in Supplementary Methods.

**Construction of expression vector of mGlu1 mutants**. Site-directed mutagenesis was performed using the QuikChange II XL site-directed mutagenesis kits (Stratagene) or the Q5® Site-Directed Mutagenesis kit (NEB) with pBluescript II SK (+) encoding rat mGlu1 (ref. [11]), according to the manufacturer's instruction. After sequencing the cDNA region of mGlu1 mutants, the cDNA was subcloned into pCAGGS vector[61] to obtain the expression vectors as previously reported[11].

**Cell culture and transfection**. HEK293 cells were maintained in Dulbecco's modified Eagle's medium (DMEM) supplemented with 100 unit ml$^{-1}$ penicillin and 100 μg ml$^{-1}$ streptomycin and 10% FBS (Sigma) at 37 °C in a humidified atmosphere of 95% air and 5% CO$_2$. HEK293 cells were transfected with plasmids (WT mGlu1, the mGlu1 mutants, or the control vector) using SuperFect transfection reagent (Qiagen) or Viafect (Promega) according to the manufacturer's instruction. The cells were co-transfected with pEGFP-F (Clontech), pDsRed monomer-F (Clontech), or piRFP670-N1 (ref. [62]) as the transfection marker. After transfection, the cells were cultured in DMEM supplemented with 10% dialyzed FBS (Gibco) instead of 10% FBS to decrease cytotoxicty. After 36–48 h incubation, the transfected HEK293 cells were utilized for experiments. piRFP670-N1 is a gift from Vladislav Verkhusha (Addgene plasmid # 45457).

PC12 cells were maintained in DMEM supplemented with 100 unit ml$^{-1}$ penicillin and 100 μg ml$^{-1}$ streptomycin and 10% horse serum (Gibco) at 37 °C in a humidified atmosphere of 95% air and 5% CO$_2$.

**Fluorescent Ca$^{2+}$ imaging**. The transfected HEK293 cells were seeded on glass coverslips (Matsunami) coated with poly-L-lysine solution (Sigma-Aldrich) then incubated for 4–10 h. HEK293 cells or cultured neurons were loaded with 5 μM Fura-2 AM (Dojindo) for 15–20 min in the growth medium. Fura-2 fluorescence was measured in HBS (107 mM NaCl, 6 mM KCl, 1.2 mM MgSO$_4$, 2 mM CaCl$_2$, 11.5 mM glucose, and 20 mM HEPES, pH 7.4). Fluorescence images were obtained using a fluorescence microscope (IX71, Olympus) equipped with a complementary metal-oxide semiconductor (CMOS) camera (ORCA-flash 4.0, Hamamatsu Photonics) under xenon-lamp illumination, and analyzed with a video imaging system (AQUACOSMOS, Hamamatsu Photonics) according to the manufacturer's protocol. The ratio of 340:380 nm fluorescence was determined from the images, on a pixel-by-pixel basis. To facilitate the screening assay in HEK293 cells, three different cell lines that expressed one of the constructs were co-cultured on a glass coverslip. Each mutant can be distinguished by co-transfected fluorescent proteins that have distinct colors as a marker, and the glutamate responses of three different mutants were assayed simultaneously. The Δratio was defined as the difference between the maximum and the initial ratio values. The Δratio was fitted with KaleidaGraph using following equation (1): a + (b-a)/(1 + (x/c)^d), and the EC$_{50}$ value was calculated.

**Western blotting analyses of HEK293 cells expressing mGlu1**. The transfected HEK293 cells were washed with PBS, and then lysed with RIPA buffer (25 mM Tris-HCl, 150 mM NaCl, pH 7.4 supplemented with 1% NP-40, 0.25% sodium deoxycholate, and 0.1% SDS) containing 1% Protease Inhibitor Cocktail (Nacalai tesque) at 4 °C for 30 min. The lysate was mixed with 5× Laemmli buffer (325 mM Tris-HCl, 15% SDS, 20% sucrose, 0.06% bromophenol blue, and 250 mM DTT, pH 6.8) and incubated at room temperature for 1 h. The sample was subject to SDS-polyacrylamide gel electrophoresis (SDS-PAGE) (7.5%) and then transferred to a polyvinylidene fluoride membrane (Bio-rad). After blocking in TBS-T (TBS with 0.05% Tween-20) supplemented with 5% skim milk powder (Wako) at room temperature for 1 h, the membranes were incubated overnight with primary antibodies in TBS-T supplemented with 1% skim milk powder at 4 °C. The membranes were washed three times with TBS-T, incubated with secondary antibodies in TBS-T with 1% skim milk powder at room temperature for 1 h, and washed three times with TBS-T. Used primary antibodies were as follows: anti-mGluR1 antibody (BD biosciences, 610964, 1:2000) and anti-β actin antibody (MBL, M177-3, 1:3000). Used secondary antibodies were as follows: anti-mouse-IgG-HRP antibody (MBL, 330, 1:3000). The signal was developed by ECL prime (Cytiva) and detected with a Fusion Solo S imaging system (Vilber Lourmat). Evolution Capt software (Vilber Lourmat fusion solo S) was used for image acquisition and analyses of blots. The target bands were manually selected, and the intensity was calculated with subtraction of background.

**Confocal live cell imaging of cell-surface mGlu1 in HEK293 cells**. HEK293 cells were co-transfected with mGlu1 plasmid (WT mGlu1, the mGlu1 mutant, or the control vector) and piRFP670-N1 as a transfection marker. The cells were added with 1 μM FITM-Cy3, and incubated for 30 min at 16 °C to suppress endocytosis. Confocal live imaging was performed with a confocal microscope (LSM900, Carl Zeiss) equipped with a 63×, numerical aperture (NA) = 1.4 oil-immersion objective. Fluorescence images were acquired by excitation at 561 or 640 nm derived from diode lasers. To quantify the fluorescence intensity of Cy3 at the cell surface, iRFP 670 positive cells were selected and the maximum signal intensity of the cell surface was calculated by ZEN blue software (Carl Zeiss). The membrane intensity was fitted with KaleidaGraph (Synergy Software), and the K$_d$ value of FITM-Cy3 was calculated.

**Evaluation of Neurite outgrowth in differentiated PC12 cells**. PC12 cells were seeded on the 24-well plate at 1.0 × 10$^4$ cells. Twelve h after seeding, the PC12 cells were differentiated by treating with 100 ng/mL NGF-β (Sigma-Aldrich), and each concentration of Pd(bpy) was added simultaneously in DMEM supplemented with 100 unit ml$^{-1}$ penicillin and 100 μg ml$^{-1}$ streptomycin and 2% FBS. After incubation for 3 days, PC12 cells bearing neurites were randomly selected, and the length of the longest neurite in each cell was measured using ImageJ package Fiji.

**Cell proliferation assay of PC12 cells**. PC12 cells were seeded on the 24-well plate at 1 × 10$^4$ cells. One to two days after seeding, the cells were treated with Pd(bpy) or Pd(sulfo-bpy) for 3-4 days. To visualize the live cells, the cells were loaded with 2 μM calcein-AM (dojindo) for 15 min in HBS, and then washed with HBS. The fluorescent images of the cells were obtained with fluorescent microscopy (Axio Observer7, Carl Zeiss) equipped with CMOS camera (Axiocam712mono, Carl Zeiss), and the cell numbers were automatically counted with ZEN blue software (Carl Zeiss).

**Animal experiments**. Pregnant ICR mice maintained under specific pathogen-free conditions and C57BL/6J mice were purchased from Japan SLC, Inc (Shizuoka, Japan). The animals were housed in a controlled environment (12 h light/dark cycle at 25 °C with 40–60% humidity) and had free access to food and water, according to the regulations of the Guidance for Proper Conduct of Animal Experiments by the Ministry of Education, Culture, Sports, Science, and Technology of Japan. All experimental procedures were performed in accordance with the National Institute of Health Guide for the Care and Use of Laboratory Animals, and were approved by the Institutional Animal Use Committees of Nagoya University, Kyoto University, Keio University, and Tsukuba University.

**Preparation of primary cortical neuronal culture**. Glass coverslips (Matsunami) in 24-well plates (BD Falcon) were coated with poly-D-lysine (Sigma-Aldrich) and washed with sterile water three times. Cerebral cortices from 16-day-old ICR mouse embryos were aseptically dissected and digested with 0.25 w/v% trypsin (Nacalai tesque) for 20 min at 37 °C. The cells were resuspended in Neurobasal Plus medium (Invitrogen) supplemented with 10% FBS, penicillin (100 units/ml), and streptomycin (100 μg/ml) and filtered by Cell Strainer (100 μm, Falcon) and centrifuged at 1,000 rpm for 5 min. The cells were resuspended in Neurobasal Plus medium supplemented with 2% of B-27 Plus Supplement (Invitrogen), 1.25 mM GlutaMAX I (Invitrogen), penicillin (100 units/ml), and streptomycin (100 μg/ml) and plated at a density of $2 \times 10^5$ cells on the 24-well plate. The cultures were maintained at 37 °C in a 95% air and 5% $CO_2$ humidified incubator. The neurons were used for fluorescent $Ca^{2+}$ imaging experiment at 3 DIV (day in vitro).

**Evaluation of the solubility of Pd complexes in ACSF**. Pd complexes were dissolved in ACSF (125 mM NaCl, 2.5 mM KCl, 2 mM $Ca_2Cl$, 1 mM $MgCl_2$, 1.25 mM $NaH_2PO_4$, 26 mM $NaHCO_3$, 10 mM Glucose) gassed with 5% $CO_2$ and 95% $O_2$, and incubated for 1 h at 4 °C. The solution was centrifuged for 15 min at 15,000 rpm, and the supernatant was collected. The UV-vis absorption spectra of the samples were measured in the range of 250–450 nm with 1 nm resolution using a UV-vis spectrophotometer UV-2600 (SHIMADZU) at room temperature. Data acquisition and analysis were performed using UVprobe software (SHIMADZU). Maximal absorption in aqueous solution at 312 nm was used to quantification.

**Generation of $mGlu1^{CBC/CBC}$ and $mGlu1^{-/-}$ by CRISPR/Cas9 system**. Mutant mice carrying mGlu1 N264H mutation were generated using the CRISPR-Cas9 system according to a previous report[22] with some modification. For generation of the mutant mice, pX330 plasmid DNA vector[63], donor single-stranded oligodeoxynucleotide (ssODN), and fertilized eggs of C57BL/6J mice were used. The sequence (5′-GACAAAATCTACAGCAATGC-3′) in exon 3 was selected as the gRNA target, and it was inserted into entry site of the pX330. The donor ssODN was designed to induce a point mutation of N264H (c.790 A > C) as shown in Fig. 5b, and the 113 mer ssDNA was synthesized (Integrated DNA Technologies). Female C57BL/6J mice were injected with pregnant mare serum gonadotropin and human chorionic gonadotropin with a 48-h interval, and mated with male C57BL/6J mice. We then collected zygotes from oviducts in mated female and mixture of the pX330 (circular, 5 ng/μL, each) and the ssODN (10 ng/μL) was microinjected into zygotes. Subsequently, survived injected zygotes were transferred into oviducts in pseudopregnant ICR female and newborns were obtained. Genotypes of the F0 mice were determined by PCR with the following primer sets; forward for WT and N264H (5′-GCAATACCACCCTCCTCTGA-3′), reverse for WT (5′-CGCATGG-CACTCAGTAACC-3′), reverse for N264H (5′- AGCTCTTCTCGCCAGCATG-3′) to give 321 bp or 218 bp product for WT or N264H, respectively. The candidate F0 mice were mated with the C57BL/6J mice to obtain F1 offspring. Genotypes of F1 mice were determined by PCR with the following primer sets; forward (5′-GCAATACCACCCTCCTCTGA-3′) and reverse (5′-CGCATGGCACTCAGT AACC-3′), and the PCR products were sequenced by Sanger sequencing. The F1 mice having mGlu1 N264H mutation were crossed with the C57BL/6J mice to obtain $mGlu1^{CBC/+}$ mice. The $mGlu1^{CBC/+}$ mice were mated to obtain $mGlu1^{CBC/CBC}$ mice.

The genotype analysis of the F1 mice also gave a mouse showing 5 bp deletion (c.790_794del) in the same exon. This deletion caused frameshift mutation and introduction of a stop codon in the same exon as shown in Supplementary Fig. 15. Thus, the indel mice were crossed with the C57BL/6J mice to obtain $mGlu1^{-/+}$ mice. The $mGlu1^{-/+}$ mice were mated to obtain $mGlu1^{-/-}$ mice.

**Preparation of acute cerebellar slices**. The mice (aged 3−5 weeks) were anesthetized with isoflurane, and their brains were transferred to ice-cold cutting solution buffer (120 mM Choline Chloride, 3 mM KCl, 8 mM $MgCl_2$, 1.25 mM $NaH_2PO_4$, 28 mM $NaHCO_3$, 22 mM Glucose, and 0.5 mM Na ascorbate) gassed with 5% $CO_2$ and 95% $O_2$ and incubated for 5 min. Sagittal cerebellar slices were cut with a microslicer (Linear slicer Pro7; DOSAKA EM) in cutting solution buffer and then incubated in ACSF gassed with 5% $CO_2$ and 95% $O_2$ at room temperature.

**Western blotting analysis of the cerebellum**. The cerebellums isolated from $mGlu1^{+/+}$ mice or $mGlu1^{CBC/CBC}$ mice, were homogenized in homogenization buffer (20 mM Tris-HCl, 150 mM NaCl, 5 mM EDTA, pH 8.0) and sonicated. Samples were solubilized by 1.0 % SDS and denatured at 70 °C in Laemmli sample buffer. Proteins were then resolved on SDS-PAGE and analyzed using western blotting with primary antibodies (polyclonal antibodies obtained from Frontier institute; anti-mGluR1a [Frontier Institute, rabbit, 1:5000], anti-GluD2 [Frontier Institute, guinea pig, 1:1000] and anti-pan AMPAR [Frontier Institute, guinea pig, 1:1000] and secondary antibodies (horseradish peroxidase-conjugated anti-rabbit antibody [Cytiva, 1:2500], or anti-guinea pig antibody [Millipore, 1:5000]). Chemiluminescent signals were detected with iBright FL1000 (Thermofisher) and quantified using ImageJ (NIH).

**Confocal live cell imaging of cell-surface mGlu1 in the cerebellar slices**. The cerebellar slices prepared from $mGlu1^{+/+}$ or $mGlu1^{CBC/CBC}$ mice, were incubated with 10 nM FITM-Cy3 and 10 μg/mL Hoechst33342 for 30 min at room temperature in a humidified atmosphere of 5% $CO_2$ and 95% $O_2$. As a negative control, the cerebellar slices were incubated with 10 nM FITM-Cy3, 10 μg/mL Hoechst33342, and 500 nM FITM. Then, the slices were subjected to confocal live imaging using a confocal microscope (LSM900, Carl Zeiss). The fluorescence intensity of Cy3 at the molecular layer were quantified by ZEN blue software (Carl Zeiss).

**Immunohistochemistry of the cerebellar slices**. Under deep anesthesia with a pentobarbital, mice were fixed by cardiac perfusion with 0.1 M sodium phosphate buffer (PB, pH 7.4) containing 4% paraformaldehyde (4% PFA/PB); the brain was removed and soaked in 4% PFA/PB for 2 h. After rinsing the specimens with PB, parasagittal slices (50-μm thickness) were prepared using a microslicer (DTK-2000; D.S.K.). Microslicer sections were permeabilized with 0.1–0.2% Triton X-100 (Sigma) in PB with 10% normal donkey serum or 2% normal goat serum/2% bovine serum albumin (BSA) for 20 min. Sections of freshly frozen brains were also prepared by cryostat (Leica) at 20 μm thickness, mounted on glass slides and fixed with 3% glyoxal for 15 min, followed by blocking with 10% donkey serum for 30 min at RT and permeabilization with PBS containing 0.1% Triton X-100 (wash buffer). Immunohistochemical staining was performed using selective primary antibodies (anti-mGlu1α [1:1000; Frontier Institute], anti-calbindin [1:10,000; Swant], anti-GluD2 [1:500; Frontier Institute], anti-Bassoon [1:500; Enzo], anti-Car8 [1:500, Frontier Institute], anti-PV [1:500; Frontier Institute], anti-GFP [1:500; Millipore] and anti-HA [1:500; Cell Signaling] antibodies) overnight at room temperature, followed by incubation with the proper fluorescent dye-conjugate secondary antibodies including anti-rabbit IgG Alexa Fluor 488, anti-rabbit IgG Cy3, anti-guinea pig IgG Alexa Fluor 488, anti-guinea pig IgG DyLight 405, ant-chiken IgY Alexa Fluor 488 and anti-goat IgG Alexa Fluor Plus 647 (1:1000, Invitrogen or Jackson ImmunoResearch) for 1 h. Finally, the sections were mounted with Fluoromount-G (SouthernBiotech). Photographs were taken with a confocal laser-scanning microscope (FLUOVIEW, FV1000; Olympus) and a super-resolution microscopy (OSR; Olympus).

**Electrophysiology of the cerebellar slices**. Whole-cell patch-clamp recordings were made from visually identified Purkinje cells using a 60× water-immersion objective attached to an upright microscope (BX51WI, Olympus) at room temperature, as described previously[64]. The resistance of patch pipettes was 1.5–3 MΩ when filled with an intracellular solution of the following composition: 65 mM Cs-methanesulfonate, 65 mM K-gluconate, 20 mM HEPES, 10 mM KCl, 1 mM $MgCl_2$, 4 mM $Na_2ATP$, 1 mM $Na_2GTP$, 5 mM sucrose, and 0.4 mM EGTA, pH 7.25 (295 mOsm/kg). The solution used for slice storage and recording consisted of the following: 125 mM NaCl, 2.5 mM KCl, 2 mM $CaCl_2$, 1 mM $MgCl_2$, 1.25 mM $NaH_2PO_4$, 26 mM $NaHCO_3$, and 10 mM D-glucose, bubbled continuously with a mixture of 95% $O_2$ and 5% $CO_2$. Picrotoxin (100 μM; Sigma) was always added to the saline to block inhibitory synaptic transmission.

To evoke EPSCs derived from CF (CF-EPSCs) and PF (PF-EPSCs) inputs onto Purkinje cells, square pulses were applied through a stimulating electrode placed on the granular layer and the molecular layer (50 μm away from the pial surface), respectively. The selective stimulation of CFs and PFs was confirmed by the paired-pulse depression (PPD) and paired-pulse facilitation (PPF) of EPSC amplitudes at a 50-ms interstimulus interval (ISI), respectively. To search for multiple CFs innervating the recorded Purkinje cell, the stimulating electrode was moved systematically in the granular layer, and the stimulation intensity was gradually increased at each stimulation site (pulse width, 50 μs; strength, 0–200 μA).

For the LTD sessions, PF-EPSCs were recorded successively at a frequency of 0.1 Hz from Purkinje cells clamped at −80 mV. After a stable PF-EPSC amplitude had been observed for at least 5 min, a conjunctive stimulation that consisted of 30 single PF stimuli together with a 200 ms depolarizing pulse from a holding potential of −60 to +20 mV was applied for LTD induction.

To chemically induce LTD (chemLTD), 50 μM (S)-3,5-dihydroxyphenylglycine (DHPG) or 1 μM Pd(sulfo-bpy) was applied for 10 min in a current-clamp mode. We often observed a transient decrease in PF-EPSC amplitude after the application of each drug probably because Purkinje cells under current-clamp conditions excite

spontaneously to suppress presynaptic function through the retrograde signaling[65]. Access resistances were monitored every 10 s by measuring the peak currents in response to 2 mV, 50-ms hyperpolarizing steps throughout the experiments; the measurements were discarded if the resistance changed by >20% of its original value. The normalized EPSC amplitude on the ordinate represents the EPSC amplitude for the average of six traces for 1 min divided by that of the average of six traces for 1 min immediately before the conjunctive stimulation or the drug application.

To monitor the excitability of MLIs, sAP-mediated current responses were extracellulary recorded by a loose cell-attached voltage-clamp at a holding potential of 0 mV. Glass electrodes used for cell-attached recordings had resistances of 3–5 MΩ when filled with ACSF. The current recordings from AAV-infected cells were performed 1−2 weeks after infection. The EPSCs and sAP-mediated currents were recorded using an Axopatch 200B amplifier (Molecular Devices), and the pClamp system (version 9.2; Molecular Devices) was used for data acquisition and analysis. The signals were filtered at 1 kHz and digitized at 4 kHz for the evoked EPSCs and 10 kHz for the sAP-mediated currents.

**Construction of AAV vectors**. To obtain pAAV-hSyn-GFP-IRES-tTA vector, DNA fragment of GFP-IRES-tTA from pAAV-TRE-fDIO-GFP-IRES-tTA[66] was inserted into the pAAV-hSyn-DIO-hM3D(Gq)-mCherry[67] lacking DIO-hM3D(Gq)-mCherry region by SalI and EcoRV digestion using the NEBuilder HiFi DNA assembly (NEB). DNA fragment of the hSyn promoter regions was exchanged for the L7 minimal promoter[68] to construct pAAV-L7-GFP-IRES-tTA.

To obtain AAV-TRE-HA-mGlu1(WT) or AAV-TRE-HA-mGlu1(N264H), DNA fragment of TRE promoter and WPRE3[35] were obtained from pAAV-TRE-fDIO-GFP-IRES-tTA[66], and SV40 late polyA was from pCI-neo vector (Promega). DNA fragment of HA-tagged mGlu1 was obtained by inserting HA-tag after the membrane localization signal of mGlu1. These DNA fragments were inserted into pAAV-hSyn-DIO-hM3D(Gq)-mCherry[67] lacking DNA fragment from hSyn promoter to hGH polyA by MluI and PmlI digestion using the NEBuilder HiFi DNA assembly (NEB). See Supplementary Fig. 14a for illustration of these vector constructs. pAAV-hSyn-DIO-hM3D(Gq)-mCherry and pAAV-TRE-fDIO-GFP-IRES-tTA are gifts from Bryan Roth (Addgene plasmid #44361) and Minmin Luo (Addgene plasmid #118026), respectively.

**Recombinant AAV production and purification**. AAVs were produced as previously described[69]. Briefly, 293AAV cells were transfected with pAdDeltaF6, pUCmini-iCAP-PHP.eB[34], and AAV vectors using polyethyleneimine "Max" (Polysciences Inc., 24765-1). In all, 72 h after transfection, cells were solubilized by freeze and thaw cycles and treated with benzonase nuclease (Millipore, 70664-3). Then viruses contained in cell lysates were purified by iodixanol gradient ultra-centrifugation (Optiprep; Abbott, AXS-1114542-250ML). Purified viral solutions were concentrated using Amicon Ultra 100 kDa (Millipore UFC910024). Viral titers were measured by quantitative PCR (Thermo Fisher, QuantStudio 5). pAd-DeltaF6 and pUCmini-iCAP-PHP.eB are gifts from James M. Wilson (Addgene plasmid #112867) and Viviana Gradinaru (Addgene plasmid #103005), respectively.

**Viral injection into live mice**. Injection of viral solutions was performed as described previously with slight modifications[70]. Briefly, under deep anesthesia, with an intraperitoneal injection of medetomidine (0.3 mg/kg), midazolam (4.0 mg/kg), and butorphanol (5.0 mg/kg), 4 µL of the solutions containing AAV mixture were directly injected into the vermis of cerebellar lobules V-VIII (200-300 µm in depth, 12 µL/h in rate) of WT mice (C57BL/6J, P20-P30) using glass pipette (30–40 µm in diameter) and microinjector (Nanoliter; World Precision Instruments). In all, 1−2 week after injection, the mice were used for each experiment.

**Statistics and reproducibility**. All graphs were generated using Microsoft Excel or KaleidaGraph. All data are expressed as mean ± s.e.m. We accumulated the data for each condition from at least three independent experiments. We evaluated statistical significance with the two-tailed Welch's t-test, two-sided Mann–Whitney U test, one-way ANOVA with Dunnet's test, two-way repeated ANOVA, or Kruskal-Wallis test followed by the Scheffe post hoc test. A value of $P < 0.05$ was considered significant. Each experiment was repeated at least three times using independent biological samples.

**Reporting summary**. Further information on research design is available in the Nature Research Reporting Summary linked to this article.

## Data availability
The authors declare that the data supporting the findings of this study are available with the paper and its Supplementary information files. The data that support the findings of this study are available from the corresponding author upon reasonable request. Source data are provided with this paper.

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

## Acknowledgements

The authors thank Dr. Maya Kono, Junko Motohashi, and Kanta Hasegawa for technical supports, and Dr. Andrew Dingley from Edanz (https://jp.edanz.com) for editing a draft of this manuscript. The pCAGGS vector was provided by the RIKEN BRC through the National Bio-Resource Project of the MEXT, Japan. This work was funded by Grants-in-Aid for Scientific Research (KAKENHI) (Grant Number 18J22952 to K.O., 20H04716, 20H03420 to W.K., 20H05628 to M.Y., 17H06348 to I.H., 16H03290, 19H05778, 20H03195 and 20H02877 to S.K.), Daiichi Sankyo Foundation of Life Science (to S.K.), the Takeda Science Foundation (to W.K and S.K.), the Mochida Memorial Foundation for Medical and Pharmaceutical Research (to W.K and S.K.), the Sumitomo Foundation (to W.K.) and the Life Science Foundation of JAPAN (to W.K.), and supported by JST CREST (JPMJCR1854) to M.Y. and JST ERATO (JPMJER1802) to I.H.

## Author contributions

S.K. and I.H. initiated and designed the project. K.O., Y. Michibata, and R.K. performed the construction of mGlu1 plasmids. K.O. and Y. Miura performed Ca$^{2+}$ imaging in HEK293 cells. K.O. and T.D. performed synthesis and confocal imaging in HEK293 cells. K.O. performed toxicity assay and solubility assay in PC12 cells. S.M. and S.T. generated mutant mice using CRISPR/Cas9 system. K.O., W.K., H.N., and S.K. bred mutant mice. K.O., W.K., T.Y., E.M., and M.Y. performed immunohistochemistry, imaging or western blot analysis in cerebellar slices. K.O., T.Y., and Y.Miura performed AAV experiments. W.K., M.I., and M.Y. performed electrophygiological experiments. K.O., W.K. M.Y., and S.K. wrote the manuscript. All authors discussed and commented on the manuscript.

## Competing interests

The authors declare no competing interests.
