## [Peer Review File · Nature Communications]

Coordination chemogenetics for activation of GPCR-type glutamate receptors in brain tissueREVIEWER COMMENTS

Reviewer #1 (Remarks to the Author):

In this manuscript, Ojima, Kakegawa and colleagues developed a chemo-genetic strategy to manipulate endogenous metabotropic mGlu1 receptors with very high pharmacological specificity. Specifically, they have developed a single-point mutation in mGlu1 (N264H) that enables orthogonal activation of the receptor with metal complexes (namely Pd(bpy)), without disturbing activation by the endogenous neurotransmitter glutamate. They then develop a metal complex analogue (Pd(sulfo-bpy)) with improved water solubility and much reduced toxicity, for use in neural preparations. Finally, they generate a knock-in (KI) mouse in which WT mGlu1 is genetically replaced by the N264H mutant. They show that this mutant mouse is phenotypically and physiologically indistinguishable from its WT counterpart, indicating that the mutant receptor is functional and expressed at endogenous sites and at endogenous levels. They then demonstrate pharmacologically-specific activation of mGlu1 in cerebellar brain slices, and chemogenetic induction of long-term depression.

Overall, this is a very impressive manuscript that introduces a novel and useful chemogenetic tool for acute and specific activation of mGlu1. The manuscript is remarkable for the wide range of techniques that are put together, from detailed molecular engineering to thorough characterization of the new tool using fluorescence imaging and a newly-developed probe, to the design and synthesis of novel metal complexes that are compatible with use on live cells, to the development of a KI mouse and its detailed characterization (notably using behavioral analysis, super resolution microscopy and slice physiology), and finally to the demonstration of the utility of the new technology using electrophysiology in brain slices. Overall the manuscript is very well written and the results are clear and convincing. The technology is potentially very useful for understanding the role of mGlu1 in brain function and in disease, by providing subtype-specific activation of mGlu1 without affecting other receptors. I only have minor issues.

- In the introduction, when the authors mention other chemogenetic strategies for manipulating receptors, they state that "Although powerful for manipulating cellular signals, these methods are not suitable for understanding the physiological roles of each receptor subtype because engineered receptors lacking affinity to endogenous ligands need to be ectopically expressed." While this is true for DREADDs and PSAM/PSEMs (they are indeed "orthogonal" for their cognate neurotransmitter ligands, because they are designed such way), this is not true for photoswitchable tethered ligands. Photoswitchable tethered ligands are anchored to the receptor surface, outside the ligand binding pocket, through introduction of a cysteine mutation (or alternatively a SNAP tag) that does not (in virtually all cases) affect the function of the receptor. Photoswitchable tethered ligands are thus very useful for understanding the endogenous functions of neurotransmitter receptors, as is the new chemogenetic strategy presented in this manuscript. This should be corrected in the text, both in the introduction and discussion. I recently wrote a review on this topic (Cell-Specific Neuropharmacology. Mondoloni S, Durand-de Cuttoli R, Mourot A., Trends Pharmacol Sci. 2019 Sep;40(9):696-710. doi: 10.1016/j.tips.2019.07.007.) which does not have to be cited in this manuscript, but may help authors update their introduction. I take this opportunity to apologize for not citing in my

review the author's 2016 Nat. Chem. paper, which is an important contribution to the field, that nevertheless escaped my radar.

- I agree with the fact that, in Fig. 1g, the EC50 for glutamate is unaffected for the double mutant A59H/N264H. However, the maximal glutamate response seems to be increased when Pd(bpy) is co-applied with glutamate (not sure whether this is statistically significant or not). If this is the case, then Pd(bpy) may also be considered as a positive allosteric modulator (PAM) for this mutant receptor. Indeed, some PAMs enhance the functional response of receptors (increased efficacy) without increase the apparent affinity for the agonist. This should be more clearly stated in the text.

- In relation with the previous comment, how do the kinetics of activation/deactivation of the mutant N264H compare between Glu and Pd(bpy)? It seems important to show that these kinetics are not fundamentally different between the two agonists.

- It remains unclear whether mGlu1-induced slow EPSCs are mediated by TRPC current alone and/or by GluD2 current as well (see for instance Ady et al. EMBO reports 2014). Text should be corrected p13.

Alexandre Mourot

Reviewer #2 (Remarks to the Author):

Ojima et al describe the discovery and characterization of a mutant mGlu1 receptor, mGlu1(N264H), and a chemogenetic method for activating this receptor using Pd complexes. The authors demonstrate that activation of this mGlu1 mutant is sufficient to evoke changes in synaptic plasticity in cerebellum in slice electrophysiology experiments using a knockin mouse. This is a well written manuscript. The experiments are performed in a methodical manner and are presented coherently. The work also somewhat advances knowledge regarding structural contributions to mGlu1 function. However, whereas the authors made a knockin mouse expressing the specific chemogenetic receptor, enthusiasm for the work is dampened by the lack of demonstration of an AAV-mediated approach to show the potential promise in utility of this chemogenetic receptor as a general chemogenetic tool. Otherwise, the chemogenetic technology described herein is limited in terms of its generalizable utility.

Certainly the use of Fura-2 is sufficient for screening and perhaps for testing but have the authors considered evaluating the performance of the function of this chemogenetic receptor more upstream such the G-protein level and not just in regard to calcium as the readout?

Fig. 5 legend., "fiver" should be "fiber"

Fig. 5., how many mice were used in these experiments? "2-5 biologically independent" is vague.

Fig. 5. Panel D, why did the authors not test FITM effects in CBC mice?

Fig. 6., how many animals were used in these experiments? Can the authors please clarify why Pd leads to a transient decrease in EPSC amplitude in the WT mice?

Reviewer #3 (Remarks to the Author):

In this study, the authors developed a chemogenetic method to activate metabotropic glutamate receptor 1 (mGlu1) directly by palladium complexes. Through a systematic screening, they identified a mGlu1 mutant, mGlu1(N264H), that was directly activated by palladium complexes and was responded normally to glutamate. Furthermore, palladium complexes had no potentiating action on the responsiveness of mGlu1(N264H) to glutamate. Then they generated mGlu1(N264H) knock-in mice to confirm that mGlu1(N264H) can be activated by palladium complexes in the mouse brain. They found no perceptible abnormality in mGlu1(N264H) knock-in mice suggesting that mGlu1(N264H) responded to the endogenous ligand glutamate normally in the mouse brain. Then they scrutinized synaptic transmission and synaptic plasticity in cerebellar slices. They found normal subcellular localization of mGlu1(N264H), normal excitatory transmission at parallel fiber to Purkinje cell and at climbing fiber to Purkinje cell synapses, normal climbing fiber synapse pruning during postnatal developmental and normal long-term depression at parallel fiber to Purkinje cell synapses. Notably, bath application of a palladium complex (Pd(sulfo-bpy)) induced LTD in homozygous and heterozygous mGlu1(N264H) knock-in mice but not in wild-type mice, suggesting that Pd(sulfo-bpy) selectively activated mGlu1(N264H) in the cerebellar slices and induced LTD.

In general, the experiments are well performed with high standard techniques and the data are convincing. I have no specific comments on experimental procedures and data. However, I am concerned that there are no new findings in this paper regarding the functions of mGluR1. The authors generated conventional global knock-in mice of mGlu1(N264H) and therefore it was impossible to perform cell type-specific activation of mGlu1(N264H) by the palladium complex. To discover a new role of mGlu1 by using the chemogenetic method of mGlu1 activation by the palladium complex, I feel it necessary to generate mice that express mGlu1(N264H) specifically in a certain cell type. In addition to the cell type-specific chemogenetic mGlu1 activation, a loss of function analysis of mGlu1 for the same cell type should be performed.

Response to Reviewer 1's comments

General Comments:

Overall, this is a very impressive manuscript that introduces a novel and useful chemogenetic tool for acute and specific activation of mGlu1. The manuscript is remarkable for the wide range of techniques that are put together, from detailed molecular engineering to thorough characterization of the new tool using fluorescence imaging and a newly-developed probe, to the design and synthesis of novel metal complexes that are compatible with use on live cells, to the development of a KI mouse and its detailed characterization (notably using behavioral analysis, super resolution microscopy and slice physiology), and finally to the demonstration of the utility of the new technology using electrophysiology in brain slices. Overall the manuscript is very well written and the results are clear and convincing. The technology is potentially very useful for understanding the role of mGlu1 in brain function and in disease, by providing subtype-specific activation of mGlu1 without affecting other receptors. I only have minor issues.

Our response

We would like to thank this reviewer for his/her kind review and for important comments. According to the suggestions and comments, we have carefully amended our manuscript as shown below. All the revisions we made are highlighted in RED in the revised manuscript.

Comment 1

- In the introduction, when the authors mention other chemogenetic strategies for manipulating receptors, they state that "Although powerful for manipulating cellular signals, these methods are not suitable for understanding the physiological roles of each receptor subtype because engineered receptors lacking affinity to endogenous ligands need to be ectopically expressed." While this is true for DREADDs and PSAM/PSEMs (they are indeed "orthogonal" for their cognate neurotransmitter ligands, because they are designed such way), this is not true for photoswitchable tethered ligands. Photoswitchable tethered ligands are anchored to the receptor surface, outside the ligand binding pocket, through introduction of a cysteine mutation (or alternatively a SNAP tag) that does not (in virtually all cases) affect the function of the receptor. Photoswitchable tethered ligands are thus very useful for understanding the endogenous functions of neurotransmitter receptors, as is the new chemogenetic strategy presented in this manuscript. This should be corrected in the text, both in the introduction and discussion. I recently wrote a review on this topic (Cell-Specific Neuropharmacology. Mondoloni S, Durand-de Cuttoli R, Mourot A., Trends Pharmacol Sci. 2019 Sep;40(9):696-710. doi: 10.1016/j.tips.2019.07.007.) which does not have to be cited in this manuscript, but may help authors update their introduction. I take this opportunity to apologize for not citing in my review the author's 2016 Nat. Chem. paper, which is an important contribution to the field, that nevertheless escaped my radar.

Our response

We agree that tethering the photoswitchable ligands is a powerful approach for understanding the endogenous functions of receptors. Thus, as suggested by this reviewer, we have changed the description of the chemogenetic strategy, especially for photoswitchable tethered ligands in both the INTRODUCTION and DISCUSSION. We also cited an appropriate review paper described by Mondoloni *et al.* as ref. #44 in the DISCUSSION.

Modification in the main text

INTRODUCTION

From page 3, line 26 to page 4, line 3: For ligand-gated ion channels, engineered receptors (PSAMs) and designed agonists (PSEMs) pairs have been reported using the bump-and-hole strategy^{9,10}. Although powerful for manipulating cellular signals, these methods are unsuitable for characterizing the physiological roles of each receptor subtype because engineered receptors lacking affinity to endogenous ligands need to be ectopically expressed.

DISCUSSION

In page 19, lines 8–25: In addition to well-used chemogenetic methods such as DREADD⁶⁻⁸ and PSAM/PSEM^{9,10}, some potential approaches for selective activation of target receptors have been achieved by chemical biology manipulations^{43,44}. For example, metal coordination has been used for functional switching of cell-surface receptors, in which the ligand-binding region of GPCRs has been mutated into the metal coordination site for metal-induced activation^{45,46}. However, native ligand-binding is affected in most cases. As an alternative, tethering photo-switchable ligands to the target protein has been demonstrated successfully via cysteine modification^{47,48} or a self-labeling enzyme tag^{48,49}, in which mutations were introduced outside or in regions distal from the ligand-binding site. Thus, native ligand-binding properties are not affected, which enabled photo-switching of particular types of neurotransmitter receptors, including glutamate receptors. Notably, photo-switching of endogenous ion-channel-type γ -aminobutyric acid A (GABA_A) receptors has been demonstrated using mice where the corresponding cysteine mutation was introduced into the genomic region encoding the GABA_A receptor⁵⁰. However, chemogenetic control of endogenous receptors is still very challenging. In this context, the current dA-CBC method, which enables control of endogenous mGlu1 function in cerebellar slices prepared from mGlu1(N264H)-knock-in mice, is a notable chemogenetic strategy for controlling endogenous receptor function.

Comment 2

- I agree with the fact that, in Fig. 1g, the EC50 for glutamate is unaffected for the double mutant A59H/N264H. However, the maximal glutamate response seems to be increased when Pd(bpy) is co-applied with glutamate (not sure whether this is statistically significant or not). If this is the case, then Pd(bpy) may also be considered as a positive allosteric modulator (PAM) for this mutant receptor. Indeed, some PAMs enhance the functional response of receptors (increased efficacy) without increase the apparent affinity for the agonist. This should be more clearly stated in the text.

Our response

Thank you for this comment. In **Figure 1g**, the Δ ratio value at 100 μ M glutamate was not significantly different ($P = 0.0664$) between Pd(bpy) (+) and Pd(bpy) (–) in the double mutant (A59H/N264H). This indicates that Pd(bpy) does not enhance the glutamate-induced responses to the mutant. Consistently, the 100 μ M glutamate-induced response was not enhanced by Pd(bpy) in the mGlu1(N264H) mutant ($P = 0.3803$) (**Figure 3c**). These indicate that Pd(bpy) does not increase the efficacy of glutamate-induced responses in these mGlu1 mutants.

Comment 3

- In relation with the previous comment, how do the kinetics of activation/deactivation of the mutant N264H compare between Glu and Pd(bpy)? It seems important to show that these kinetics are not fundamentally different between the two agonists.

Our response

As suggested by this reviewer, we evaluated both activation and deactivation kinetics of Pd(bpy)-induced activation in the N264H mutant. Regarding the activation kinetics, the time required for Pd(bpy)-induced mGlu1 activation to reach a peak ratio (58.0 ± 8.5 sec) was longer than that of the glutamate-induced activation (25.7 ± 1.9 sec) (**Supplementary Fig. 6a, b**). We also evaluated the wash-out of Pd(bpy) to estimate the deactivation kinetics. The wash-out process of Pd(bpy) (49.0 ± 6.4 sec) was similarly slower than that of glutamate (22.0 ± 3.2 sec) (**Supplementary Fig. 6a, c**). These indicate that the kinetics is slightly different between Pd(bpy) and glutamate. However, we believe that this difference is not so problematic for chemogenetic modulation, because mGlu1 contributes to slow excitatory transmission in neurons. Importantly, repeated ratio increases were observed after washing out Pd(bpy) in the culture medium (**Fig. 3g**), showing the reversible action of Pd(bpy) for direct activation of mGlu1.

Modification in the main text

RESULTS

In page 9, lines 1–8: The time required for Pd(bpy)-induced mGlu1 activation to reach a peak ratio was longer when compared with that of glutamate-induced activation (58.0 ± 8.5 s or 25.7 ± 1.9 s for Pd(bpy) or glutamate, respectively) (**Supplementary Fig. 6**). The kinetics of the wash-out of Pd(bpy) was also slower than that of glutamate (49.0 ± 6.4 s or 22.0 ± 3.2 s for Pd(bpy) or glutamate, respectively). These observations indicate that the binding kinetics are slightly different between Pd(bpy) and glutamate. However, repeated ratio increases were observed after washing out Pd(bpy) in the culture medium (**Fig. 3g**), showing the reversible action of Pd(bpy) for direct activation of mGlu1.

Modification in Supplementary information

Supplementary Figure 6 entitled “The kinetics of Pd(bpy)-induced responses in the mGlu1(N264H) mutant” has been newly added.

Comment 4

- It remains unclear whether mGlu1-induced slow EPSCs are mediated by TRPC current alone and/or by GluD2 current as well (see for instance Ady et al. EMBO reports 2014). Text should be corrected p13.

Our response

As suggested by this reviewer, we modified the main text by citing the corresponding references.

Modification in the main text

RESULTS

In page 13, lines 10–14: Furthermore, burst PF stimulation (2–10 times of PF stimuli at 100 Hz) induced robust **TRPC- or TRPC/GluD2-mediated slow currents**^{29,30} designated as slowEPSCs by activation of the perisynaptic mGlu1 in both *mGlu1^{CBC/CBC}* and *mGlu1^{+/+}* Purkinje cells (**Fig. 5j**, *P* = 0.898 by two-way repeated ANOVA).

Response to Reviewer 2's comments

General Comments:

Ojima et al describe the discovery and characterization of a mutant mGlu1 receptor, mGlu1(N264H), and a chemogenetic method for activating this receptor using Pd complexes. The authors demonstrate that activation of this mGlu1 mutant is sufficient to evoke changes in synaptic plasticity in cerebellum in slice electrophysiology experiments using a knockin mouse. This is a well written manuscript. The experiments are performed in a methodical manner and are presented coherently. The work also somewhat advances knowledge regarding structural contributions to mGlu1 function. However, whereas the authors made a knockin mouse expressing the specific chemogenetic receptor, enthusiasm for the work is dampened by the lack of demonstration of an AAV-mediated approach to show the potential promise in utility of this chemogenetic receptor as a general chemogenetic tool. Otherwise, the chemogenetic technology described herein is limited in terms of its generalizable utility.

Our response

We would like to thank this reviewer for his/her kind review and important comments. According to the suggestions and comments, we have carefully amended our manuscript. All the revisions we made are highlighted in RED in the revised manuscript.

To respond to this comment, we examined the applicability of dA-CBC method for cell-type-specific activation of the mGlu1 mutant in cerebellar slices using AAV vectors having cell-type-specific promoters. Using the AAV-mediated system, we successfully expressed the mGlu1 N264H mutant in Purkinje cells using AAV having L7 promoter (**Fig. 7b, c**). Notably, Pd(sulfo-bpy)-mediated chemLTD was observed from the infected cells (**Fig. 7d**). These indicate that AAV-mediated dA-CBC strategy works well for cell-type-specific activation of mGlu1 in cerebellar slices.

Previous studies have revealed that mGlu1 is highly expressed in molecular layer interneurons (MLIs) as well as Purkinje cells in the cerebellum. In addition, bath application of DHPG, a mGlu1 activator, has increased the spontaneous action potential (sAP) in MLIs (Refs. 36, 37). However, it is not completely clear whether this effect was mediated by direct activation of mGlu1 expressed in MLIs since Ca²⁺ increases in Purkinje cells could retrogradely activate MLIs by dendritic release of glutamate (Ref. 38). To clarify this point, the mGlu1 N264H mutant was selectively expressed in MLIs using AAV having the hSyn promoter (**Fig. 7f**). Notably, Pd(sulfo-bpy) increased the firing rate of MLIs expressed with the mGlu1 mutant effectively (**Fig. 7g**). In contrast, Pd(sulfo-bpy) failed to increase the firing rate of MLIs surrounded by mGlu1(N264H)-expressing Purkinje cells. These results indicate that mGlu1 in MLIs, but not in Purkinje cells, directly regulates the excitability of MLIs. Taken together, cell-type-specific activation of mGlu1 was demonstrated successfully using the dA-CBC strategy and AAV techniques.

Overall, in the revised manuscript, we added the new section entitled “Cell-type-specific activation of mGlu1 in the cerebellum using dA-CBC” in RESULTS with new **Figure 7** and **Supplementary Figure 14**. In addition, a new paragraph for cell-type-specific dA-CBC strategy is added in the DISCUSSION. We believe that these data and descriptions strongly support the usefulness

of our chemogenetic approach. Thanks again for your valuable comments.

Modification in the main text

ABSTRACT

In page 1, lines 12–15: Moreover, cell-type-specific activation of mGlu1 was demonstrated successfully using adeno-associated viruses, which shows the potential utility of this chemogenetics for clarifying the physiological roles of mGlu1 in a cell-type-specific manner.

INTRODUCTION

In page 4, lines 24–27: Moreover, the dA-CBC strategy was applied successfully to cell-type-specific activation of mGlu1 using adeno-associated viruses (AAVs) encoding the mGlu1 mutant. Thus, dA-CBC represents a powerful method for understanding the physiological roles of mGlu1 in brain tissue.

RESULTS

From page 15, line 7 to page 17, line 21: **Cell-type-specific activation of mGlu1 in the cerebellum using dA-CBC.** mGlu1 is widely expressed not only in the cerebellum but also in various brain regions such as the olfactory bulb, thalamus, and hippocampus²⁴. Therefore, to understand the roles of mGlu1 in each brain region, a cell- or region-type-specific mGlu1 activation system is desirable. For this purpose, we examined the applicability of the dA-CBC strategy for cell-type-specific activation of mGlu1 in cerebellar slices using adeno-associated viruses (AAVs) with each cell-type-specific promoter (L7 or human synapsin (hSyn) for Purkinje cells or ML interneurons (MLIs), respectively). Considering the strict packaging capacity limit (5.2 kb) of the AAV and the large transgene size (3.6 kb) of mGlu1 cDNA, we applied two-component AAVs using the tetracycline (tet)-inducible system — a tet-off transactivator (tTA) and a tet-responsive element (TRE)³⁴ (**Fig. 7a** and **Supplementary Fig. 14a**). In this system, Purkinje cell-specific expression of tTA should be achieved using an AAV vector (pAAV-L7-GFP-IRES-tTA) with the L7 promoter (0.8 kb), in which GFP is co-expressed under the same promoter as a transfection marker. A second AAV vector encodes mGlu1 under a short TRE promoter (0.3 kb) (pAAV-TRE-HA-mGlu1 WT or pAAV-TRE-HA-mGlu1(N264H)), in which a short regulatory element (WPRES and SV40 polyA (0.4 kb))³⁵ is used instead of the conventional regulatory element (WPRES and bGH polyA (1.1 kb)). We expected that these shorter elements allow acceptance of the large mGlu1 cDNA even in the packaging capacity limit of AAVs.

For selective expression of mGlu1(N264H) in Purkinje cells, a mixture of AAV-TRE-HA-mGlu1(N264H) and AAV-L7-GFP-IRES-tTA, termed AAV-L7-mGlu1^{CBC}, was injected directly into the cerebellum of WT mice, and brains were fixed and sliced 8 days after viral infection. After immunostaining the slices, the immunofluorescent signals of GFP merged well with the anti-carbonic anhydrase VIII (Car8) signals, a Purkinje cell-specific marker (**Fig. 7b**). Immunostaining using an anti-HA antibody indicated that HA-tagged mGlu1(N264H) was detected in GFP-positive cells. Consistently, western blotting using the anti-HA antibody revealed that the molecular weight of the transgene product corresponds to that of mGlu1 (**Fig. 7c**). In western blotting analyses, the anti-mGlu1 signals in the AAV-infected mice were stronger than those in non-infected mice. These results indicate that mGlu1(N264H) was abundantly expressed in a Purkinje cells-specific manner using the AAVs. In addition, similar results were obtained for mice infected with a mixture of AAV-TRE-mGlu1(WT) and AAV-L7-GFP-IRES-tTA, termed AAV-L7-mGlu1^{WT} (**Supplementary Fig. 14b**).

To examine whether the AAV-mediated dA-CBC strategy works, we next recorded chemLTD using acutely prepared cerebellar slices from WT mice infected with AAV-L7-mGlu1^{CBC}. As shown in **Figure 7d**, bath application of 1 μ M Pd(sulfo-bpy) prominently induced chemLTD in the GFP-positive Purkinje cells ($66 \pm 5\%$ at $t = 31\text{--}35$ min), which is consistent with the results using *mGlu1*^{CBC/CBC} or *mGlu1*^{CBC/+} mice (**Fig. 6d**). In contrast, Pd(sulfo-bpy)-mediated chemLTD was not observed in the GFP-positive Purkinje cells of the slices infected with AAV-L7-mGlu1^{WT} ($88 \pm 7\%$ at $t = 31\text{--}35$ min, $P = 0.047$ by Mann-Whitney U test; **Fig. 7d**). These observations indicate that the AAV-mediated dA-CBC strategy works well for cell-type-specific activation of mGlu1 in cerebellar slices.

Finally, to establish the potential utility of the AAV-mediated dA-CBC strategy, mGlu1(N264H) was expressed in MLIs of the cerebellum, where mGlu1 is highly expressed^{36,37} (**Fig. 7e**), using the AAV vector with the hSyn promoter. Eight days after co-injection of AAV-TRE-mGlu1(N264H) with AAV-hSyn-GFP-IRES-tTA, termed AAV-hSyn-mGlu1^{CBC}, into the cerebellum of WT mice, the immunofluorescent signals of GFP were detected primarily from MLIs (Car8-negative and parvalbumin (PV)-positive cells) in the cerebellar slice (**Fig. 7f**). The GFP signals merged well with the anti-HA signals, indicating the selective expression of mGlu1(N264H) in MLIs (**Fig. 7f** and **Supplementary Fig. 14c**). Importantly, in MLIs, several groups have reported that activation of mGlu1 increases the excitability in MLIs, which was revealed by bath application of the mGlu1 activator, DHPG^{36,37}. However, it is not completely clear whether this effect is mediated by the direct activation of mGlu1 expressed in MLIs since $[Ca^{2+}]_i$ increases in Purkinje cells could retrogradely activate MLIs by the dendritic release of glutamate³⁸. To clarify this point, we examined selective activation of mGlu1(N264H) in MLIs using our system. Consistent with previous reports^{36,37}, bath application of DHPG clearly increased the number of spontaneous action potentials (sAP) in both AAV-hSyn-mGlu1^{CBC} and AAV-hSyn-mGlu1^{WT}-infected MLIs (**Supplementary Fig. 14d**). Notably, as shown in **Figure 7g**, 10 μ M of Pd(sulfo-bpy) increased the firing rate of MLIs infected with AAV-hSyn-mGlu1^{CBC} effectively ($P = 4.613 \times 10^{-5}$ by Wilcoxon signed-rank test) but not MLIs infected with AAV-hSyn-mGlu1^{WT} ($P = 0.453$ by Wilcoxon signed-rank test). In contrast, Pd(sulfo-bpy) failed to increase the firing rate of MLIs surrounded by mGlu1(N264H)-expressing Purkinje cells that were infected with AAV-L7-mGlu1^{CBC} ($P = 0.293$ by Wilcoxon signed-rank test; **Fig. 7g**). These results indicate that mGlu1 in MLIs, but not in Purkinje cells, directly regulates the excitability of MLIs. Taken together, cell-type-specific activation of mGlu1 was demonstrated successfully using the dA-CBC strategy and AAV techniques.

DISCUSSION

In page 18, lines 11–14: Moreover, dA-CBC was applied successfully for Purkinje cells- or MLI-specific activation of mGlu1 using AAVs, which shows the potential utility of this chemogenetic technique for understanding the physiological roles of mGlu1 in a cell-type-specific manner.

From page 19, line 26 to page 20, line 17: Furthermore, a cell-type-specific dA-CBC method was established by combining this method with the AAV expression system. Using this strategy, we successfully activated mGlu1 expression in Purkinje cells or MLIs independently in the cerebellum. Although mGlu1 in Purkinje cells has been studied extensively by analyzing Purkinje cell-specific mGlu1 transgenic rescue mice^{39,40}, physiological roles of mGlu1 in MLIs are largely unknown. Bath application of DHPG reportedly enhances cellular excitability of MLIs in acute cerebellar slices^{36,37}; however, it remains unclear which cell

(i.e., Purkinje cells or MLIs) or which type (i.e., mGlu1 or mGlu5) of mGlu is responsible for the event. By using a cell-type-specific dA-CBC method, we found that mGlu1 in MLIs directly regulates the excitability of neurons. Although mGlu-dependent excitation has been observed in different brain regions or different types of neurons, such as hippocampal CA1 pyramidal neurons, striatal dopaminergic neurons and retinal ganglion cells, the downstream signals that induce neuronal depolarization and inward currents are unique in each cell-type^{13,51,52}. Among them, delta-type glutamate receptors (GluDs) are reported to open the gate following mGlu1 activation in Purkinje cells and striatal dopaminergic neurons⁵¹. Interestingly, GluDs are enriched in MLIs and Purkinje cells in the cerebellum⁵³. Future efforts will use our cell-type-specific dA-CBC method to determine which downstream signals regulate MLI excitability. Thus, the cell-type-specific dA-CBC strategy is a powerful and general tool for mGlu1 activation in the brain.

Modification in Figure and Supplementary information

Figure 7 entitled “Cell-type-specific activation of mGlu1 in the cerebellum using dA-CBC and AAVs” has been newly added.

Supplementary Figure 14 entitled “AAV-mediated expression of mGlu1 in the cerebellum for dA-CBC” has been newly added.

Comment 1

Certainly the use of Fura-2 is sufficient for screening and perhaps for testing but have the authors considered evaluating the performance of the function of this chemogenetic receptor more upstream such the G-protein level and not just in regard to calcium as the readout?

Our response

To confirm the selective action of Pd(bpy) on the Gq-pathway, we examined the effect of YM-254890, a Gq protein-selective inhibitor. As shown in **Supplementary Figure 7c**, Pd(bpy)-induced responses were significantly suppressed by the pretreatment of YM-254890, supporting the action of Pd(bpy) on the Gq-protein pathway. In addition, we also examined IP3 production by Pd(bpy) as another readout as suggested by this reviewer. In this experiment, IP3 production was evaluated by the accumulation of inositol 1-phosphate (IP1), the degradation product of IP3. As shown in **Supplementary Figure 8**, IP1 production was enhanced by Pd(bpy) or glutamate in the mGlu1 N264H mutant. This also supports Pd(bpy) activates the mGlu1-dependent Gq pathway.

Modification in the main text

RESULTS

In page 9, lines 9–19: We also examined the effects of Pd(bpy) on mGlu1-dependent Gq signaling. The Pd(bpy)-induced Ca²⁺ responses were inhibited by LY367385 or FITM, a competitive or non-competitive antagonist of mGlu1, respectively (**Supplementary Fig. 7a, b**). The Ca²⁺ responses were also suppressed by YM-254890, a Gq-selective inhibitor, and U73122, a PLC inhibitor (**Supplementary Fig. 7c, d**), indicating that Pd(bpy) activates the conventional Gq-pathway of mGlu1. Consistently, inositol 1,4,5-triphosphate (IP3) production was increased by Pd(bpy) in HEK293 cells transfected with mGlu1(N264H), which was evaluated by the accumulation of inositol 1-phosphate (IP1), the degradation product of IP3. (**Supplementary Fig. 8**).

Thus, we can directly and reversibly activate mGlu1 and its downstream signal pathways with minimal disturbance to receptor function by using Pd(bpy) and a single-point mutation (N264H) of mGlu1 in the dA-CBC method.

Modification in Supplementary information

Supplementary Figure 7c has been newly added.

Supplementary Figure 8 entitled “Measurement of IP1 accumulation via Gq-pathway” has been newly added.

Comment 2

Fig. 5 legend., “fiver” should be “fiber”

Our response

We corrected the error, as suggested by this reviewer.

Modification in Figure

Figure 5a legend has been corrected.

Comment 3

Fig. 5., how many mice were used in these experiments? “2-5 biologically independent” is vague.

Our response

As suggested by this reviewer, we wrote the exact number of mice in the Figure legend. In addition, we increased the number of mice in this revision.

Modification in Figure

Figure 5d legend has been modified.

Comment 4

Fig. 5. Panel D, why did the authors not test FITM effects in CBC mice?

Our response

We newly examined FITM effects on the *mGlu1^{CBC/CBC}* mice. This data is added in **Figure 4d**. Because of the limited space, we moved fluorescent images including Hoechst dye-staining to Supplementary information (**Supplementary Fig. 10b**).

Modification in Figure and Supplementary information

Figure 7d has been modified.

Supplementary Figure 10d has been newly added.

Comment 5

Fig. 6., how many animals were used in these experiments? Can the authors please clarify why Pd leads

to a transient decrease in EPSC amplitude in the WT mice?

Our response

We added the description of the number of animals to the **Figure 6** legend. The transient decrease in the EPSC amplitude is often observed after switching between the current- and voltage-clamp modes in wild-type Purkinje cells independently of LTD. For example, the transient decrease in the EPSC amplitude was recorded in *Clql1*-null or *Bai3*-null Purkinje cells lacking LTD (see figures below). The reason is not completely clear, but we postulate that the endocannabinoid-mediated retrograde signaling is activated by transient depolarization of Purkinje cells by switching the clamp mode (Ref. 65). Thus, we believe this is unrelated to the action of Pd(sulfo-bpy). We have included a brief description of this phenomenon in the METHODS of the revised manuscript.

Kakegawa et al., *Neuron* 85, 316 (2015)

Modification in the main text

METHODS

In page 31, lines 2–5: We often observed a transient decrease in PF-EPSC amplitude after the application of each drug probably because Purkinje cells under current-clamp conditions excite spontaneously to suppress presynaptic function through the retrograde signaling⁶⁵.

Response to Reviewer 3's comments

In this study, the authors developed a chemogenetic method to activate metabotropic glutamate receptor 1 (mGlu1) directly by palladium complexes. Through a systematic screening, they identified a mGlu1 mutant, mGlu1(N264H), that was directly activated by palladium complexes and was responded normally to glutamate. Furthermore, palladium complexes had no potentiating action on the responsiveness of mGlu1(N264H) to glutamate. Then they generated mGlu1(N264H) knock-in mice to confirm that mGlu1(N264H) can be activated by palladium complexes in the mouse brain. They found no perceptible abnormality in mGlu1(N264H) knock-in mice suggesting that mGlu1(N264H) responded to the endogenous ligand glutamate normally in the mouse brain. Then they scrutinized synaptic transmission and synaptic plasticity in cerebellar slices. They found normal subcellular localization of mGlu1(N264H), normal excitatory transmission at parallel fiber to Purkinje cell and at climbing fiber to Purkinje cell synapses, normal climbing fiber synapse pruning during postnatal developmental and normal long-term depression at parallel fiber to Purkinje cell synapses. Notably, bath application of a palladium complex (Pd(sulfo-bpy)) induced LTD in homozygous and heterozygous mGlu1(N264H) knock-in mice but not in wild-type mice, suggesting that Pd(sulfo-bpy) selectively activated mGlu1(N264H) in the cerebellar slices and induced LTD.

In general, the experiments are well performed with high standard techniques and the data are convincing. I have no specific comments on experimental procedures and data. However, I am concerned that there are no new findings in this paper regarding the functions of mGluR1. The authors generated conventional global knock-in mice of mGlu1(N264H) and therefore it was impossible to perform cell type-specific activation of mGlu1(N264H) by the palladium complex. To discover a new role of mGlu1 by using the chemogenetic method of mGlu1 activation by the palladium complex, I feel it necessary to generate mice that express mGlu1(N264H) specifically in a certain cell type. In addition to the cell type-specific chemogenetic mGlu1 activation, a loss of function analysis of mGlu1 for the same cell type should be performed.

Our response

We would like to thank this reviewer for his/her kind review and important comments. According to the suggestions and comments, we have carefully amended our manuscript. All the revisions we made are highlighted in RED in the revised manuscript.

To respond to this reviewer, we demonstrated cell-type-specific expression of the mGlu1(N264H) mutant in live mice using AAVs vectors having cell-type-specific promoters. Using the AAV-mediated system, we successfully expressed the mGlu1(N264H) mutant in Purkinje cells using AAV having L7 promoter (**Fig. 7b, c**). Notably, Pd(sulfo-bpy)-mediated chemLTD was observed from the infected cells (**Fig. 7d**). These indicate that AAV-mediated dA-CBC strategy works well for cell-type-specific activation of mGlu1 in cerebellar slices.

We also examined the applicability of the dA-CBC method for activating mGlu1 in a cell-type-specific manner to clarify the cell-type-specific roles of mGlu1. Previous studies have revealed that mGlu1 is highly expressed in molecular layer interneurons (MLIs) as well as Purkinje cells in the

cerebellum. Importantly, in MLIs, several groups have reported that activation of mGlu1 increased the spontaneous action potential (sAP) in the neuron, which was revealed by bath application of DHPG, a mGlu1 activator (Refs. 36, 37). However, it is not completely clear whether this effect is mediated by the direct activation of mGlu1 expressed in MLIs since Ca^{2+} increases in Purkinje cells could retrogradely activate MLIs by the dendritic release of glutamate (Ref. 38). To clarify this point, the mGlu1(N264H) mutant was selectively expressed in MLIs using AAV having the hSyn promoter (**Fig. 7f**). Notably, Pd(sulfo-bpy) effectively increased the firing rate of MLIs expressed with the mGlu1 mutant (**Fig. 7g**). In contrast, Pd(sulfo-bpy) failed to increase the firing rate of MLIs surrounded by mGlu1(N264H)-expressing Purkinje cells. These results indicate that mGlu1 in MLIs but not in Purkinje cells directly regulates the excitability of MLIs.

Overall, we successfully demonstrated cell-type-specific activation of mGlu1, which clarified cell-type-specific roles of mGlu1 in the cerebellum. As suggested by this reviewer, loss of function analyses of mGlu1 for the same cell type would be required to confirm the physiological roles of mGlu1. However, considering that the main claim of this paper is the development of a new chemogenetic technique for mGlu1, we would like to focus on the developing dA-CBC technique and its applicability in this paper. Alternatively, we have added one paragraph regarding cell-type-specific roles of mGlu1 in the DISCUSSION. Thanks again for your valuable comments.

Modification in the main text

ABSTRACT

In page 1, lines 12–15: **Moreover, cell-type-specific activation of mGlu1 was demonstrated successfully using adeno-associated viruses, which shows the potential utility of this chemogenetics for clarifying the physiological roles of mGlu1 in a cell-type-specific manner.**

INTRODUCTION

In page 4, lines 24–27: **Moreover, the dA-CBC strategy was applied successfully to cell-type-specific activation of mGlu1 using adeno-associated viruses (AAVs) encoding the mGlu1 mutant. Thus, dA-CBC represents a powerful method for understanding the physiological roles of mGlu1 in brain tissue.**

RESULTS

From page 15, line 7 to page 17, line 21: **Cell-type-specific activation of mGlu1 in the cerebellum using dA-CBC.** mGlu1 is widely expressed not only in the cerebellum but also in various brain regions such as the olfactory bulb, thalamus, and hippocampus²⁴. Therefore, to understand the roles of mGlu1 in each brain region, a cell- or region-type-specific mGlu1 activation system is desirable. For this purpose, we examined the applicability of the dA-CBC strategy for cell-type-specific activation of mGlu1 in cerebellar slices using adeno-associated viruses (AAVs) with each cell-type-specific promoter (L7 or human synapsin (hSyn) for Purkinje cells or ML interneurons (MLIs), respectively). Considering the strict packaging capacity limit (5.2 kb) of the AAV and the large transgene size (3.6 kb) of mGlu1 cDNA, we applied two-component AAVs using the tetracycline (tet)-inducible system — a tet-off transactivator (tTA) and a tet-responsive element (TRE)³⁴ (**Fig. 7a** and **Supplementary Fig. 14a**). In this system, Purkinje cell-specific expression of tTA should be achieved using an AAV vector (pAAV-L7-GFP-IRES-tTA) with the L7 promoter (0.8 kb), in which GFP is co-expressed under the same promoter as a transfection marker. A second AAV vector encodes mGlu1

under a short TRE promoter (0.3 kb) (pAAV-TRE-HA-mGlu1 WT or pAAV-TRE-HA-mGlu1(N264H)), in which a short regulatory element (WPRE3 and SV40 polyA (0.4 kb))³⁵ is used instead of the conventional regulatory element (WPRE and bGH polyA (1.1 kb)). We expected that these shorter elements allow acceptance of the large mGlu1 cDNA even in the packaging capacity limit of AAVs.

For selective expression of mGlu1(N264H) in Purkinje cells, a mixture of AAV-TRE-HA-mGlu1(N264H) and AAV-L7-GFP-IRES-tTA, termed AAV-L7-mGlu1^{CBC}, was injected directly into the cerebellum of WT mice, and brains were fixed and sliced 8 days after viral infection. After immunostaining the slices, the immunofluorescent signals of GFP merged well with the anti-carbonic anhydrase VIII (Car8) signals, a Purkinje cell-specific marker (**Fig. 7b**). Immunostaining using an anti-HA antibody indicated that HA-tagged mGlu1(N264H) was detected in GFP-positive cells. Consistently, western blotting using the anti-HA antibody revealed that the molecular weight of the transgene product corresponds to that of mGlu1 (**Fig. 7c**). In western blotting analyses, the anti-mGlu1 signals in the AAV-infected mice were stronger than those in non-infected mice. These results indicate that mGlu1(N264H) was abundantly expressed in a Purkinje cells-specific manner using the AAVs. In addition, similar results were obtained for mice infected with a mixture of AAV-TRE-mGlu1(WT) and AAV-L7-GFP-IRES-tTA, termed AAV-L7-mGlu1^{WT} (**Supplementary Fig. 14b**).

To examine whether the AAV-mediated dA-CBC strategy works, we next recorded chemLTD using acutely prepared cerebellar slices from WT mice infected with AAV-L7-mGlu1^{CBC}. As shown in **Figure 7d**, bath application of 1 μ M Pd(sulfo-bpy) prominently induced chemLTD in the GFP-positive Purkinje cells ($66 \pm 5\%$ at $t = 31-35$ min), which is consistent with the results using *mGlu1^{CBC/CBC}* or *mGlu1^{CBC/+}* mice (**Fig. 6d**). In contrast, Pd(sulfo-bpy)-mediated chemLTD was not observed in the GFP-positive Purkinje cells of the slices infected with AAV-L7-mGlu1^{WT} ($88 \pm 7\%$ at $t = 31-35$ min, $P = 0.047$ by Mann-Whitney U test; **Fig. 7d**). These observations indicate that the AAV-mediated dA-CBC strategy works well for cell-type-specific activation of mGlu1 in cerebellar slices.

Finally, to establish the potential utility of the AAV-mediated dA-CBC strategy, mGlu1(N264H) was expressed in MLIs of the cerebellum, where mGlu1 is highly expressed^{36,37} (**Fig. 7e**), using the AAV vector with the hSyn promoter. Eight days after co-injection of AAV-TRE-mGlu1(N264H) with AAV-hSyn-GFP-IRES-tTA, termed AAV-hSyn-mGlu1^{CBC}, into the cerebellum of WT mice, the immunofluorescent signals of GFP were detected primarily from MLIs (Car8-negative and parvalbumin (PV)-positive cells) in the cerebellar slice (**Fig. 7f**). The GFP signals merged well with the anti-HA signals, indicating the selective expression of mGlu1(N264H) in MLIs (**Fig. 7f** and **Supplementary Fig. 14c**). Importantly, in MLIs, several groups have reported that activation of mGlu1 increases the excitability in MLIs, which was revealed by bath application of the mGlu1 activator, DHPG^{36,37}. However, it is not completely clear whether this effect is mediated by the direct activation of mGlu1 expressed in MLIs since $[Ca^{2+}]_i$ increases in Purkinje cells could retrogradely activate MLIs by the dendritic release of glutamate³⁸. To clarify this point, we examined selective activation of mGlu1(N264H) in MLIs using our system. Consistent with previous reports^{36,37}, bath application of DHPG clearly increased the number of spontaneous action potentials (sAP) in both AAV-hSyn-mGlu1^{CBC} and AAV-hSyn-mGlu1^{WT}-infected MLIs (**Supplementary Fig. 14d**). Notably, as shown in **Figure 7g**, 10 μ M of Pd(sulfo-bpy) increased the firing rate of MLIs infected with AAV-hSyn-mGlu1^{CBC} effectively

($P = 4.613 \times 10^{-5}$ by Wilcoxon signed-rank test) but not MLIs infected with AAV-hSyn-mGlu1^{WT} ($P = 0.453$ by Wilcoxon signed-rank test). In contrast, Pd(sulfo-bpy) failed to increase the firing rate of MLIs surrounded by mGlu1(N264H)-expressing Purkinje cells that were infected with AAV-L7-mGlu1^{CBC} ($P = 0.293$ by Wilcoxon signed-rank test; **Fig. 7g**). These results indicate that mGlu1 in MLIs, but not in Purkinje cells, directly regulates the excitability of MLIs. Taken together, cell-type-specific activation of mGlu1 was demonstrated successfully using the dA-CBC strategy and AAV techniques.

DISCUSSION

In page 18, lines 11–14: Moreover, dA-CBC was applied successfully for Purkinje cells- or MLI-specific activation of mGlu1 using AAVs, which shows the potential utility of this chemogenetic technique for understanding the physiological roles of mGlu1 in a cell-type-specific manner.

From page 19, line 26 to page 20, line 17: Furthermore, a cell-type-specific dA-CBC method was established by combining this method with the AAV expression system. Using this strategy, we successfully activated mGlu1 expression in Purkinje cells or MLIs independently in the cerebellum. Although mGlu1 in Purkinje cells has been studied extensively by analyzing Purkinje cell-specific mGlu1 transgenic rescue mice^{39,40}, physiological roles of mGlu1 in MLIs are largely unknown. Bath application of DHPG reportedly enhances cellular excitability of MLIs in acute cerebellar slices^{36,37}; however, it remains unclear which cell (i.e., Purkinje cells or MLIs) or which type (i.e., mGlu1 or mGlu5) of mGlu is responsible for the event. By using a cell-type-specific dA-CBC method, we found that mGlu1 in MLIs directly regulates the excitability of neurons. Although mGlu-dependent excitation has been observed in different brain regions or different types of neurons, such as hippocampal CA1 pyramidal neurons, striatal dopaminergic neurons and retinal ganglion cells, the downstream signals that induce neuronal depolarization and inward currents are unique in each cell-type^{13,51,52}. Among them, delta-type glutamate receptors (GluDs) are reported to open the gate following mGlu1 activation in Purkinje cells and striatal dopaminergic neurons⁵¹. Interestingly, GluDs are enriched in MLIs and Purkinje cells in the cerebellum⁵³. Future efforts will use our cell-type-specific dA-CBC method to determine which downstream signals regulate MLI excitability. Thus, the cell-type-specific dA-CBC strategy is a powerful and general tool for mGlu1 activation in the brain.

Modification in Figure and Supplementary information

Figure 7 entitled “Cell-type-specific activation of mGlu1 in the cerebellum using dA-CBC and AAVs” has been newly added.

Supplementary Figure 14 entitled “AAV-mediated expression of mGlu1 in the cerebellum for dA-CBC” has been newly added.

REVIEWERS' COMMENTS

Reviewer #1 (Remarks to the Author):

The authors have adequately addressed my requests and I have no further comment.

Reviewer #2 (Remarks to the Author):

The authors have done an excellent job addressing my comments. Thank you.

Reviewer #3 (Remarks to the Author):

In this revised version, the authors added data with AAV-mediated expression of mGlu1(N264H) specifically into Purkinje cells or MLIs of the cerebellum. They successfully induced Purkinje cell-specific expression of mGlu1(N264H) by using the L7 promoter and showed Pd(sulfo-bpy)-mediated chemLTD in mGlu1(N264H) expressing Purkinje cells. Furthermore, they induced MLI-specific expression of mGlu1(N264H) by using the hSyn promoter and revealed that the activation of mGlu1 in MLIs but not in other cell-types of the cerebellum enhanced the frequency of spontaneous firing of MLIs. These are clear examples of the results from cell-type-specific activation of mGlu1(N264H) by Pd(sulfo-bpy). Although loss of function analyses are required to confirm the physiological roles of mGlu1, I understand the authors' claim that the main point of this paper is the development of a new chemogenetic technique for mGlu1.